# Interpretable Reward Redistribution in Reinforcement Learning: A Causal Approach

**Yudi Zhang[1]  Yali Du[2]  Biwei Huang[3]  Ziyan Wang[2]  Jun Wang[4]**
**Meng Fang[5, 1]  Mykola Pechenizkiy[1]**
[1]Eindhoven University of Technology   [2]King's College London
[3]University of California San Diego   [4]University College London
[5]University of Liverpool
{y.zhang5,m.pechenizkiy}@tue.nl, {yali.du,ziyan.wang}@kcl.ac.uk
bih007@ucsd.edu, jun.wang@cs.ucl.ac.uk, Meng.Fang@liverpool.ac.uk

## Abstract

A major challenge in reinforcement learning is to determine which state-action pairs are responsible for future rewards that are delayed. Reward redistribution serves as a solution to re-assign credits for each time step from observed sequences. While the majority of current approaches construct the reward redistribution in an uninterpretable manner, we propose to explicitly model the contributions of state and action from a causal perspective, resulting in an interpretable reward redistribution and preserving policy invariance. In this paper, we start by studying the role of causal generative models in reward redistribution by characterizing the generation of Markovian rewards and trajectory-wise long-term return and further propose a framework, called *Generative Return Decomposition* (GRD), for policy optimization in delayed reward scenarios. Specifically, GRD first identifies the unobservable Markovian rewards and causal relations in the generative process. Then, GRD makes use of the identified causal generative model to form a compact representation to train policy over the most favorable subspace of the state space of the agent. Theoretically, we show that the unobservable Markovian reward function is identifiable, as well as the underlying causal structure and causal models. Experimental results show that our method outperforms state-of-the-art methods and the provided visualization further demonstrates the interpretability of our method. The project page is located at https://reedzyd.github.io/GenerativeReturnDecomposition/.

## 1 Introduction

Reinforcement Learning (RL) has achieved significant success in a variety of applications, such as autonomous driving [1, 2], robot [3, 4], games [5, 6, 7], financial trading [8, 9], and healthcare [10]. A challenge in real-world RL is the delay of reward feedback [11, 12, 13, 14]. In such a delayed reward setting, the learning process may suffer from instability due to the lack of proper guidance from the reward sequences [15, 16, 17]. Methods such as reward shaping [18, 19], curiosity-driven intrinsic reward [20, 21, 22, 23] and hindsight relabeling [24, 25, 26, 13] have been proposed to provide proxy rewards in RL.

How to compute the contribution of each state-action pair affected by a delayed reward and explain the reasons behind such contribution are equally important, as they can provide insights into the underlying dynamics and the decision process, guiding the development of more effective RL algorithms. Recent return decomposition methods employ the return equivalent hypothesis to produce proxy rewards for the state-action pair at each time step [15, 16, 17]. These methods allow for the

decomposition of the trajectory-wise return into Markovian rewards, preserving the policy invariance. However, previous methods construct reward redistribution by hand-designed rules [15, 16], or an uninterpretable model [17], where the explanation of how the state and action contribute to the proxy reward is unclear. On the other hand, recently, causal modeling has shown promise in environmental model estimation and follow-up policy learning for RL [27, 28]. Works along this direction investigate and characterize the significant causes with respect to their expected outcomes, which help improve the efficiency of exploration by making use of structural constraints [29, 30, 31], resulting in an impressive performance improvement. Therefore, causal modeling serves as a natural tool to investigate the contributions of states and actions toward the Markovian rewards.

In this paper, we propose a novel algorithm for reward redistribution with causal treatment, called Generative Return Decomposition (GRD). GRD tackles reward redistribution with causal modeling that enjoys the following advantages. First, instead of a flat representation, GRD specifies each state and action as a combination of values of several constituent variables relevant to the problem and accounts for the causal relationships among variables in the system. Such a structural factored representation, relating to Factored MDP [32, 33], provides favorability to form and identify the Markovian reward function from the perspective of causality. Unlike previous approaches, GRD utilizes such a parsimonious graphical representation to discover how each dimension of state and action contributes to the Markovian reward. Moreover, within the generative process of the MDP envi-

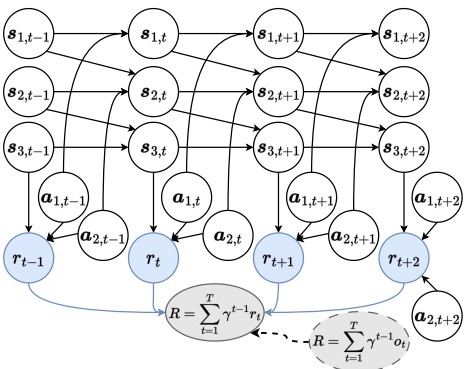

Figure 1: A graphical illustration of causal structure in Generative Return Decomposition. See the main text for the interpretation.

ronment, we can naturally explain and model the observed delayed return as a causal effect of the unobserved Markovian reward sequence, which provides insights into the underlying dynamics of the environment, preserves the policy invariance as well. Figure 1 shows the framework of GRD, involving the causal relationship among environmental variables. The nodes denote different variables in the MDP environment, $i.e.$, all dimensions of state $s_{\cdot,t}$ and action $a_{\cdot,t}$, Markovian rewards $r_t$ for $t \in [1, T]$, as well as the long-term return $R$. For sparse reward settings in RL, the Markovian rewards $r_t$ are unobservable, which are represented by nodes with blue filling. While considering the return-equivalent assumption in return decomposition, we can observe the trajectory-wise long-term return, $R$, which equals the discounted sum of delayed reward $o_t$ and evaluates the performance of the agent within the whole episode. A special case of delayed rewards is in episodic RL, where $o_{1:T-1} = 0$ and $o_T \neq 0$. Theoretically, we prove that the underlying generative process, including the unknown causal structure, the Markovian reward function, and the dynamics function, are identifiable. GRD learns the Markovian reward function and dynamics function in a component-wise way to recover the underlying causal generative process, resulting in an explainable and principled reward redistribution. Furthermore, we identify a minimal sufficient representation for policy training from the learned parameters, consisting of the dimensions of the state that have an impact on the generated Markovian rewards, both directly and indirectly. Such a compact representation has a matching causal structure with the generation of the Markovian reward, aiding in the effectiveness and stability of policy learning.

We summarize the main contributions of this paper as follows. First, we reformulate the reward redistribution by introducing a graphical representation to describe the causal relationship among the dimensions of state, action, the Markovian reward, and the long-term return. The causal structure over the environmental variables, the unknown Markovian reward function, and the dynamics function are identifiable, which is supported by theoretical proof. Second, we propose GRD to learn the underlying causal generative process. Based on the learned model, we construct interpretable reward redistribution and identify a compact representation to facilitate policy learning. Furthermore, empirical experiments on robot tasks from the MuJoCo environment demonstrate that our method outperforms the state-of-the-art methods and the provided visualization shows the interpretability of the redistributed rewards, indicating the usefulness of causal modeling for the sparse reward settings.

## 2 Related Work

Below we review the related work on reward redistribution and causality-facilitated RL methods.

Previous work redistributes rewards to deal with the delayed rewards in RL, such as reward shaping [18, 19] and curiosity-driven intrinsic reward [20, 21, 22, 23]. Return decomposition draws attention since RUDDER [15] rethinks the return-equivalent condition for reward shaping [34] to decompose the long-term return into proxy rewards for each time step through an LSTM-based long-term return predictor and manually designed assignment. Align-RUDDER [16] introduces the bioinformatical alignment method to align successful demonstration sequences, then manually scores new sequences according to the aligned demonstrations. However, in many reinforcement learning (RL) tasks, obtaining a sufficient number of successful demonstrations can be challenging. [35] and [36] explore to improve RUDDER by the replacement of expressive language models [37] and the continuous modern Hopfield network [38] to decompose delayed rewards. Apart from them, RRD [17] proposes an upper bound for the common return-equivalent assumption to serve as a surrogate optimization objective, bridging the return decomposition and uniform reward redistribution in IRCR [39]. However, those methods lack the explanation of how the contributions derives [17, 35], or applying unflexible manual design [15, 16] to decompose the long-term returns into dense proxy rewards for the collected sequences. By contrast, we study the role of the causal model, investigate the relationships within the generation of the Markovian reward and exploit them to guide interpretable return decomposition and efficient policy learning.

Plenty of work explores solving diverse RL problems with *causal treatment*. Most conduct research on the transfer ability of RL agents. For instance, [27] learns factored representation and an individual change factor for different domains, and [31] extends it to cope with non-stationary changes. More recently, [28, 40] remove unnecessary dependencies between states and actions variables in the causal dynamics model to improve the generalizing capability in the unseen state. Also, causal modeling is introduced to multi-agent task [41, 42], model-based RL [43], imitation learning [44] and so on. However, most of them do not consider the cases of MDP with observed delayed and sparse rewards explicitly. As an exception, [45] distinguishes the impact of the actions from one-time step and future time step to the delayed feedback by a policy gradient estimator, but still suffering from very delayed sparse rewards. Compared with the previous work, we investigate the causes for the generation of sequences of Markovian rewards to address very delayed rewards, even episodic delayed ones.

## 3 Preliminaries

In this section, we review the Markov Decision Process [46] and return decomposition [15, 16, 17].

**Markov Decision Process** (MDP) is represented by a tuple $\langle \mathcal{S}, \mathcal{A}, \mathcal{R}, \gamma, P \rangle$, where $\mathcal{S}$, $\mathcal{A}$, $\mathcal{R}$ denote the state space, action space, and reward function, separately. The state transition probability of the environment is denoted as $P(s_{t+1} \mid s_t, a_t)$. The goal of reinforcement learning is to find an optimal policy $\pi : \mathcal{S} \rightarrow \mathcal{A}$ to maximize the expected long-term return, *i.e.*, a discounted sum of the rewards with the predefined discounted factor $\gamma$ and episode length $T$, $J(\pi) = \mathbb{E}_{s_t \sim P(s_t|s_{t-1}, a_{t-1}), a_t \sim \pi(a_t|s_t)}[\sum_{t=1}^{T} \gamma^{t-1} \mathcal{R}(s_t, a_t)]$.

**Return decomposition** is proposed to decompose the long-term return feedback into a sequence of Markovian rewards in RL with delayed rewards [15, 16] while maintaining the invariance of optimal policy [15]. In the case of delayed reward setting, the agent can only observe some sparse delayed rewards. An extreme case is that the agent can only get an episodic non-zero reward $o_T$ at the end of each episode, *i.e.*, $o_{1:T-1} = 0$ and $o_T \neq 0$, called episodic reward [17]. In general, the observed rewards $o_t$ are delayed and usually harm policy learning, since the contributions of the state-action pairs are not clear. Therefore, return decomposition is proposed to generate proxy rewards $\hat{r}_t$ for each time step, which is expected to be non-delayed, dense, Markovian, and able to clarify the contributions of the state-action pairs to the delayed feedback. The work along this line shares a common assumption,

$$R = \sum_{t=1}^{T} \gamma^{t-1} o_t = \sum_{t=1}^{T} \gamma^{t-1} \hat{r}_t, \tag{1}$$

where $R$ is the long-term return and $o_t$, $\hat{r}_t$ denote the observed delayed reward and the decomposed proxy reward for each time step, separately.

# 4 Causal Reformulation of Reward Redistribution

As a foundation, below we introduce a Dynamic Bayesian Network (DBN) [47] to reformulate reward redistribution and characterize the underlying generative process, leading to a natural interpretation of the explicit contribution of each dimension of state and action towards the Markovian rewards.

## 4.1 Causal Modeling

We describe the MDP process with a trajectory-wise long-term return, $\mathcal{P} = \langle \mathcal{S}, \mathcal{A}, g, P_f, \gamma, \mathcal{G} \rangle$. $\mathcal{S}, \mathcal{A}$ represent the sets of states $s$ and actions $a$, respectively. $\mathcal{G}$ denotes a DBN that describes the causal relationship within the MDP environment, constructed over a finite number of random variables, $\{s_{1,t}, \cdots, s_{|s|,t}, a_{1,t}, \cdots, a_{|a|,t}, r_t\}_{t=1}^T \cup R$ where $|s|$ and $|a|$ are the dimension of $s$ and $a$. $r_t$ is the unobservable Markovian reward that serves as the objective of return decomposition. The state transition $P_f$ can be represented as, $P_f(s_{t+1} \mid s_t, a_t) = \prod_{i=1}^{|s|} P(s_{i,t+1} \mid \mathrm{pa}(s_{i,t+1}))$, where $i$ is the index of dimension of $s_t$. Here $\mathrm{pa}(s_{i,t+1})$ denotes the causal parents of $s_{i,t+1}$ and usually is a subset of the dimensions of $s_t$ and $a_t$. We assume a given initial state distribution $P(s_1)$. Similarly, we define the functions, $g$, which generates unobservable Markovian reward, $r_t = g(\mathrm{pa}(r_t))$, where $\mathrm{pa}(r_t)$ is a subset of the dimensions of $s_t$ and $a_t$. The trajectory-wise long-term return $R$ is the causal effect of all the Markovian rewards. An example of the causal structure denoted by $\mathcal{G}$ is given in Figure 1. For simplicity, we reformalize the generative process as,

$$
\begin{cases}
s_{i,t+1} = f(\boldsymbol{C}^{s \to s}_{\cdot, i} \odot \boldsymbol{s}_t, \boldsymbol{C}^{a \to s}_{\cdot, i} \odot \boldsymbol{a}_t, \epsilon_{s,i,t}) \\
r_t = g(\boldsymbol{C}^{s \to r} \odot \boldsymbol{s}_t, \boldsymbol{C}^{a \to r} \odot \boldsymbol{a}_t, \epsilon_{r,t}) \\
R = \sum_{t=1}^T \gamma^{t-1} r_t
\end{cases}
\tag{2}
$$

for $i = (1, 2, \cdots, |s|)$. $\odot$ is the element-wise product. $f$ and $g$ stand for the dynamics function and reward function, separately. $\boldsymbol{C}^{\cdot \to \cdot}$ are categorical masks organizing the causal structure in $\mathcal{G}$. $\epsilon_{s,i,t}$ and $\epsilon_{r,t}$ are i.i.d random noises. Specially, $\boldsymbol{C}^{s \to r} \in \{0,1\}^{|s|}$ and $\boldsymbol{C}^{a \to r} \in \{0,1\}^{|a|}$ control if a specific dimension of state $s_t$ and action $a_t$ impact the Markovian reward $r_t$, separately. For example, if there is an edge from $s_{i,t}$ to $r_t$, then the $\boldsymbol{C}^{s \to r}_i = 1$. Similarly, $\boldsymbol{C}^{s \to s} \in \{0,1\}^{|s| \times |s|}$ and $\boldsymbol{C}^{a \to s} \in \{0,1\}^{|a| \times |s|}$ indicate the causal relationship between $s_t$, $a_t$ and $s_{t+1}$, separately. In particular, we assume that $\mathcal{G}$ is time-invariant, $i.e.$, $f$, $g$, $\boldsymbol{C}^{\cdot \to \cdot}$ are invariant. In the system, $s_t$, $a_t$, and $R$ are observable, while $r_t$ are unobservable and there are no unobserved confounders and instantaneous causal effects.

## 4.2 Identifiability of Generative Process

Below we give the identifiability result of learning the latent causal structure and unknown functions in the above causal generative model.

**Proposition 1 (Identifiability)** *Suppose the state $s_t$, action $a_t$, trajectory-wise long-term return $R$ are observable while Markovian rewards $r_t$ are unobservable, and they form an MDP, as described in Eq. 2. Then, under the global Markov condition and faithfulness assumption, the reward function $g$ and the Markovian rewards $r_t$ are identifiable, as well as the causal structure that is characterized by binary masks $\boldsymbol{C}^{\cdot \to \cdot}$ and the transition dynamics $f$.*

Details of the proof for Proposition 1 are deferred to Appendix B. Proposition 1 provides the foundation for us to identify the causal structure $\boldsymbol{C}^{\cdot \to \cdot}$ and the unknown functions $f$, $g$ in the generative process from the observed data. As a result, with the identified structures and functions in Eq. 2, we can naturally construct an interpretable return decomposition. Here we clarify the difference between our work and the previous work for a better understanding: 1) compared with [17, 35], $g$ in Eq. 2 is a general description to characterize the Markovian reward as the causal effect of a subset of the dimensions of state and action, 2) compared with [15, 16], treating long-term reward $R$ as the causal effect of all the Markovian rewards in a sequence is an interpretation of return-equivalent assumption from the causal view, allowing a reasonable and flexible return decomposition.

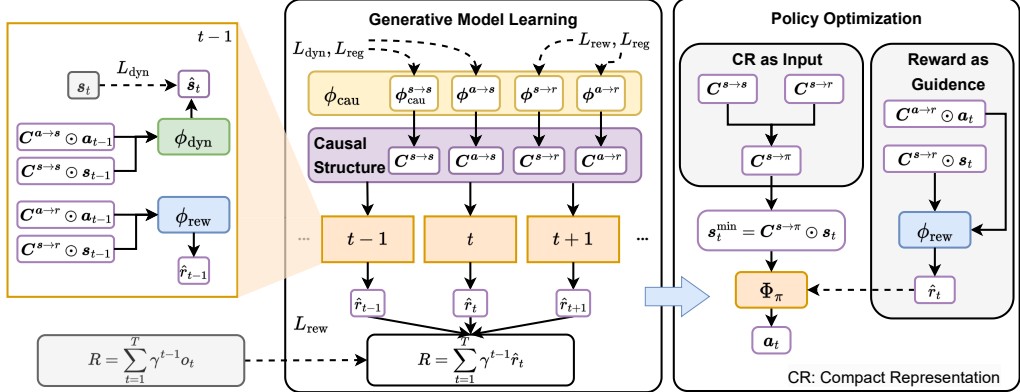

Figure 2: The framework of the proposed Generative Return Decomposition (GRD). $\phi_{\mathrm{cau}}, \phi_{\mathrm{rew}}, \phi_{\mathrm{dyn}}$ in generative model $\Phi_{\mathrm{m}}$ are marked as yellow, blue and green, while policy model $\Phi_\pi$ is marked as orange. The observable variables, state $s_t$, action $a_t$, and the delayed reward $o_t$, are marked as gray. The mediate results, binary masks, $C^{\cdot \to \cdot}$, outputs of policy, the predicted Markovian rewards $\hat{r}_t$ and the compact representation $s_t^{\min}$ are denoted as purple squares. The policy model $\Phi_\pi$ takes as input $s_t^{\min}$ and its training is supervised by the predicted Markovian reward $\hat{r}_t$. The dotted lines represent the supervision signals, *i.e.*, losses and predicted Markovian rewards.

## 5 Generative Return Decomposition

In this section, we propose a principled framework for reward redistribution from the causal perspective, named Generative Return Decomposition (GRD). Specifically, we first introduce how GRD recovers a generative model within the MDP environment in Sec. 5.1, and then show how GRD deploys the learned parameters of the generative model to facilitate efficient policy learning in Sec. 5.2. The GRD consists of two parts, $\Phi_{\mathrm{m}}$ for the parameterized generative process and $\Phi_\pi$ for the policy. Therefore, the corresponding loss function includes $L_m(\Phi_m)$ and $J_\pi(\Phi_\pi)$, for causal generative model estimation and policy optimization, respectively. Hence, the overall objective can be formulated as

$$\min_{\Phi_m, \Phi_\pi} L(\Phi_m, \Phi_\pi) = L_m(\Phi_m) + J_\pi(\Phi_\pi). \tag{3}$$

In the following subsections, we will present each component of the objective function.

### 5.1 Generative Model Estimation

In this subsection, we present how our proposed GRD recovers the underlying generative process. This includes the identification of binary masks ($C^{\cdot \to \cdot}$), as well as the estimation of unknown functions ($f$ and $g$). An overview of our method is illustrated in Figure 2. The parameterized model $\Phi_m$ that incorporates the structures and functions in Eq. 2 is used to approximate the causal generative process. The optimization is carried out by minimizing $L_{\mathrm{m}}$ over the replay buffer $\mathcal{D}$, which consists of trajectories, with each trajectory $\tau = \{\langle s_t, a_t \rangle |_{t=1}^T, R\}$, where $R$ is the discounted sum of observed delayed rewards $o_t$, as given in Eq. 1.

**Generative Model.** The parameterized model $\Phi_{\mathrm{m}}$ consists of three parts, $\phi_{\mathrm{cau}}, \phi_{\mathrm{dyn}}, \phi_{\mathrm{rew}}$, which are described below. **(1)** $\phi_{\mathrm{cau}}$ is used to identify the causal structure in $\mathcal{G}$ by predicting the values of binary masks in Eq. 2. It consists of four parts, $\phi_{\mathrm{cau}}^{s \to s} \in \mathbb{R}^{|s| \times |s| \times 2}, \phi_{\mathrm{cau}}^{a \to s} \in \mathbb{R}^{|a| \times |s| \times 2}, \phi_{\mathrm{cau}}^{s \to r} \in \mathbb{R}^{|s| \times 2}$ and $\phi_{\mathrm{cau}}^{a \to r} \in \mathbb{R}^{|a| \times 2}$, which are used to predict $C^{s \to s}, C^{a \to s}, C^{s \to r}$ and $C^{a \to r}$, separately. We take the prediction of $C^{s \to s}$ with $\phi_{\mathrm{cau}}^{s \to s}$ as an example to explain the process of predicting causal structure. $\phi_{\mathrm{cau}}^{s \to s}$ characterizes $|s| \times |s|$ i.i.d Bernoulli distributions for the existence of the edges organized in the matrix, $C^{s \to s}$. Each Bernoulli distribution is denoted by a two-element vector, where each element corresponds to the unnormalized probability of classifying the edge as existing or not, respectively. We denote the probability of the existence of an edge from $i$-th dimension of the state to $j$-th dimension of the next state in the causal graph as $P(C_{i,j}^{s \to s})$. Binary prediction of $C^{s \to s}$ is sampled by applying element-wise gumbel-softmax [48] during training, while using greedy selection during

inference. Similarly, we can obtain $P(\boldsymbol{C}_{i,j}^{\boldsymbol{a}\to\boldsymbol{s}})$, $P(\boldsymbol{C}_i^{\boldsymbol{s}\to r})$, $P(\boldsymbol{C}_i^{\boldsymbol{a}\to r})$, $\boldsymbol{C}^{\boldsymbol{a}\to\boldsymbol{s}}$, $\boldsymbol{C}^{\boldsymbol{s}\to r}$ and $\boldsymbol{C}^{\boldsymbol{a}\to r}$. **(2)** $\phi_{\text{rew}}$ is constructed with fully-connected layers and approximates the reward function $g$ in Eq. 2, which takes as input the state $\boldsymbol{s}_t$, action $\boldsymbol{a}_t$, as well as the predictions of $\boldsymbol{C}^{\boldsymbol{s}\to r}$ and $\boldsymbol{C}^{\boldsymbol{a}\to r}$ to obtain the prediction of Markovian rewards. **(3)** $\phi_{\text{dyn}}$ is constructed on a Mixture Density Network [49], and is used to approximate the dynamics function $f$ in Eq. 2, which takes as inputs the state $\boldsymbol{s}_t$, action $\boldsymbol{a}_t$, the predictions of causal structures, $\boldsymbol{C}^{\boldsymbol{s}\to\boldsymbol{s}}$ and $\boldsymbol{C}^{\boldsymbol{a}\to\boldsymbol{s}}$. As an example, the predicted distribution for the $i$-th dimension of the next state is $P(\boldsymbol{s}_{i,t+1} \mid \boldsymbol{s}_t, \boldsymbol{a}_t, \boldsymbol{C}_{\cdot,i}^{\boldsymbol{s}\to\boldsymbol{s}}, \boldsymbol{C}_{\cdot,i}^{\boldsymbol{a}\to\boldsymbol{s}}; \phi_{\text{dyn}})$. More details for $\Phi_{\text{m}}$ can be found in the Appendix C.2.

**Loss Terms.** Accordingly, the loss term $L_{\text{m}}$ contains three components with $L_{\text{m}}(\Phi_{\text{m}}) = L_{\text{rew}} + L_{\text{dyn}} + L_{\text{reg}}$. Considering the long-term return $R$ as the causal effect of the Markovian rewards $r_t$, we optimize $\phi_{\text{rew}}$, $\phi_{\text{cau}}^{\boldsymbol{s}\to r}$ and $\phi_{\text{cau}}^{\boldsymbol{a}\to r}$ by

$$L_{\text{rew}}(\phi_{\text{rew}}, \phi_{\text{cau}}^{\boldsymbol{s}\to r}, \phi_{\text{cau}}^{\boldsymbol{a}\to r}) = \mathbb{E}_{\tau\sim\mathcal{D}}[\|R - \sum_{t=1}^{T}\gamma^{t-1}\hat{r}_t\|^2] = \mathbb{E}_{\tau\sim\mathcal{D}}[\|\sum_{t=1}^{T}\gamma^{t-1}o_t - \sum_{t=1}^{T}\gamma^{t-1}\hat{r}_t\|^2],$$
(4)

where $\hat{r}_t$ is predicted by $\phi_{\text{rew}}$, *i.e.*, $\hat{r}_t = \phi_{\text{rew}}(\boldsymbol{s}_t, \boldsymbol{a}_t, \boldsymbol{C}^{\boldsymbol{s}\to r}, \boldsymbol{C}^{\boldsymbol{a}\to r})$. To optimize $\phi_{\text{dyn}}$, $\phi_{\text{cau}}^{\boldsymbol{s}\to\boldsymbol{s}}$ and $\phi_{\text{cau}}^{\boldsymbol{a}\to\boldsymbol{s}}$, we minimize

$$L_{\text{dyn}}(\phi_{\text{dyn}}, \phi_{\text{cau}}^{\boldsymbol{s}\to\boldsymbol{s}}, \phi_{\text{cau}}^{\boldsymbol{a}\to\boldsymbol{s}}) = \mathbb{E}_{\boldsymbol{s}_t, \boldsymbol{a}_t, \boldsymbol{s}_{t+1}\sim\mathcal{D}}[-\sum_{i=1}^{|\boldsymbol{s}|}\log P\left(\boldsymbol{s}_{i,t+1} \mid \boldsymbol{s}_t, \boldsymbol{a}_t, \boldsymbol{C}_{\cdot,i}^{\boldsymbol{s}\to\boldsymbol{s}}, \boldsymbol{C}_{\cdot,i}^{\boldsymbol{a}\to\boldsymbol{s}}; \phi_{\text{dyn}}\right)]. \quad (5)$$

Additionally, we minimize the following cross-entropy terms to regulate the sparsity of learned causal structure to avoid trial solutions. It is achieved by force to optimize the parameters towards the direction of the nonexistence of the causal edge. Let $D_i(\boldsymbol{x}) = \log P(\boldsymbol{x}_i)$, where $P(\boldsymbol{x}_i)$ is the possibility that the edge $\boldsymbol{x}_i$ exists. The regularizer term is,

$$L_{\text{reg}}(\phi_{\text{cau}}) = \underbrace{\lambda_1 \sum_i D_i(\boldsymbol{C}^{\boldsymbol{s}\to r})}_{\text{state-to-reward}} + \underbrace{\lambda_2 \sum_i D_i(\boldsymbol{C}^{\boldsymbol{a}\to r})}_{\text{action-to-reward}} +$$
$$\underbrace{\lambda_3 \sum_{j\neq i} D_{i,j}(\boldsymbol{C}^{\boldsymbol{s}\to\boldsymbol{s}})}_{\text{state-to-state (excluding self-connections)}} + \underbrace{\lambda_4 \sum_{j=i} D_{i,j}(\boldsymbol{C}^{\boldsymbol{s}\to\boldsymbol{s}})}_{\text{state-to-state (self-connections)}} + \underbrace{\lambda_5 \sum_{i,j} D_{i,j}(\boldsymbol{C}^{\boldsymbol{a}\to\boldsymbol{s}})}_{\text{action-to-state}}. \quad (6)$$

where $\mathbb{1}(\cdot)$ is the indicator function and hyper-parameters $\lambda_{(\cdot)}$ are listed in Appendix C.5.

## 5.2 Policy Optimisation with Generative Models

Here we explain how the learned generative model aids policy optimization with delayed rewards.

**Compact Representation as Inputs.** Inspired by [30], we identify a minimal and sufficient state set, $\boldsymbol{s}^{\text{min}}$, for policy learning as the input of the policy model, $\Phi_\pi$, called *compact representation*. It contains the states' dimensions that directly or indirectly impact the reward. To be more specific, it includes each dimension of state $\boldsymbol{s}_{i,t} \in \boldsymbol{s}_t$, which either 1) has an edge to the reward $r_t$ *i.e.*, $\boldsymbol{C}_i^{\boldsymbol{s}\to r} = 1$, or 2) has an edge to another state component in the next time-step, $\boldsymbol{s}_{j,t+1} \in \boldsymbol{s}_t$, *i.e.*, $\boldsymbol{C}_{i,j}^{\boldsymbol{s}\to\boldsymbol{s}} = 1$, such that the same component at time $t$ is in the compact representation, *i.e.*, $\boldsymbol{s}_{j,t} \in \boldsymbol{s}_t^{\text{min}}$. We first select $\boldsymbol{C}^{\boldsymbol{s}\to\boldsymbol{s}}$ and $\boldsymbol{C}^{\boldsymbol{s}\to r}$ greedily from the i.i.d Bernoulli distributions characterized by $\phi_{\text{cau}}^{\boldsymbol{s}\to\boldsymbol{s}}$ and $\phi_{\text{cau}}^{\boldsymbol{s}\to r}$ (as illustrated in Appendix C.2) and organize the state dimensions which exist in $\boldsymbol{s}_t^{\text{min}}$ as $\boldsymbol{C}^{\boldsymbol{s}\to\pi}$. We define $\mathbb{C} := \{i, \forall \boldsymbol{C}_i^{\boldsymbol{s}\to r} = 1\}$ to denote the dimensions of $\boldsymbol{s}_t$ which have impact on the Markovian reward $r_t$ directly. Then we have,

$$\boldsymbol{C}^{\boldsymbol{s}\to\pi} = \{\boldsymbol{C}^{\boldsymbol{s}\to r}\} \vee \{\boldsymbol{C}_{\cdot,\mathbb{C}_1}^{\boldsymbol{s}\to\boldsymbol{s}} \vee \cdots \vee \boldsymbol{C}_{\cdot,\mathbb{C}_{|C|}}^{\boldsymbol{s}\to\boldsymbol{s}}\}, \quad (7)$$

where $|C|$ denotes the size of $\mathbb{C}$ and $\vee$ denotes element-wise logical or operation. Then, the compact representation which serves as the input of policy is defined as, $\boldsymbol{s}_t^{\text{min}} = \boldsymbol{C}^{\boldsymbol{s}\to\pi} \odot \boldsymbol{s}_t$. In this way, GRD considers the dimensions contributing to the current reward provider (the learned Markovian reward function) to choose action, thus leading to efficient policy learning.

**Markovian Rewards as Guidance.** With the predicted Markovian rewards, we adopt soft actor-critic (SAC) [50] for policy optimization. Specifically, given $\boldsymbol{s}_t$ and $\boldsymbol{a}_t$, we replace the observed delayed reward $o_t$ by $\hat{r}_t = \phi_{\text{rew}}(\boldsymbol{s}_t, \boldsymbol{a}_t, \boldsymbol{C}^{\boldsymbol{s}\to r}, \boldsymbol{C}^{\boldsymbol{a}\to r})$, where $\boldsymbol{C}^{\boldsymbol{s}\to r}$ and $\boldsymbol{C}^{\boldsymbol{a}\to r}$ are selected greedily from $\phi_{\text{cau}}^{\boldsymbol{s}\to r}$ and $\phi_{\text{cau}}^{\boldsymbol{a}\to r}$, as illustrated in Appendix C.2. The policy model $\Phi_\pi$ contains two parts, a critic $\phi_v$

and an actor $\phi_\pi$ due to the applied SAC algorithm. SAC alternates between a policy evaluation step, which trains a critic $\phi_v(\boldsymbol{s}_t^{\min}, \boldsymbol{a}_t)$ to estimate $Q^\pi(\boldsymbol{s}_t^{\min}, \boldsymbol{a}_t) = \mathbb{E}_\pi[\sum_{t=1}^T \gamma^{t-1}\hat{r}(\boldsymbol{s}_t^{\min}, \boldsymbol{a}_t) \mid \boldsymbol{s}_1 = \boldsymbol{s}, \boldsymbol{a}_1 = \boldsymbol{a}]$ using the Bellman backup operator, and a policy improvement step, which trains an actor $\phi_\pi(\boldsymbol{s}_t^{\min})$ by minimizing the expected KL-divergence,

$$J_\pi(\Phi_\pi) = \mathbb{E}_{\boldsymbol{s}_t \sim \mathcal{D}}\left[D_{\mathrm{KL}}\left(\Phi_\pi \| \exp\left(Q^\pi - V^\pi\right)\right)\right], \tag{8}$$

where $V^\pi$ is the value function of state-action pairs [50]. More detailed implementation of $\Phi_\pi$ can be founded in Appendix C.3.

We optimize the generative model and the policy model simultaneously, cutting the need for collecting additional diverse data for causal discovery. It is worth noticing that SAC can be replaced by any policy learning backbones [51, 52, 53, 54]. Besides, for those model-based ones [53, 54], our learned generative model serves as a natural alternative to the used environment model.

## 6 Experiments

In this section, we begin by performing experiments on a collection of MuJoCo locomotion benchmark tasks to highlight the remarkable performance of our methods. We then conduct an ablation study to demonstrate the effectiveness of the compact representation. Finally, we investigate the interpretability of our GRD by visualizing learned causal structure and decomposed rewards.

### 6.1 Setup

Below we introduce the tasks, baselines, and evaluation metric in our experiments.

**Tasks.** We evaluate our method on eight widely used classical robot control tasks in the MuJoCo environment [55], including *Half-Cheetah*, *Ant*, *Walker2d*, *Humanoid*, *Swimmer*, *Hopper*, *Humanoid-Standup*, and *Reacher* tasks. All robots are limited to observing only one episode reward at the end of each episode, which is equivalent to the accumulative Markov reward that describes their performance throughout the episode. The maximum episode length is shared among all tasks and set to be $1000$. In these tasks, the contributions across dimensions in the state to the Markovian reward can differ significantly due to the agents' different objectives, even in *Humanoid* and *HumanoidStandup*, where the robots hold the same dynamic physics. A common feature is that the control costs are designed to penalize the robots if they take overly large actions. Therefore, the grounded Markovian reward functions consider different dimensions of state and all dimensions of action with varying weights.

**Baselines.** We compare our GRD with RRD (biased) [17], RRD (unbiased) [17], and IRCR [39], whose implementation details are provided in Appendix C.1. We use the same hyper-parameters for policy learning as RRD for a fair comparison. While RUDDER and Align-RUDDER are also popular methods for return decomposition, RRD and IRCR demonstrate superior performance compared to them. Moreover, Align-RUDDER requires successful trajectories to form a scoring rule for state-action pairs, which are unavailable in MuJoCo. Therefore, we do not compare our performance with theirs.

**Evaluation Metric.** We report the average accumulative reward across $5$ seeds with random initialization to demonstrate the performance of evaluated methods. Intuitively, a method that earns higher accumulative rewards in evaluation indicates better reward redistribution.

### 6.2 Main Results

Figure 3 provides an overview of the performance comparison among the full version of our method, GRD, an ablation version without compact representation, GRD w/o CR, and the baseline methods. The ablation version GRD w/o CR uses all state dimensions instead of the compact representation as the input of policy model. For better visualization, we scale the time step axis to highlight the different converging speeds for each task. Our method GRD outperforms the baseline methods in all tasks, as shown by higher average accumulative rewards and faster convergence, especially in the tasks with high-dimensional states. For instance, in *HalfCheetah* and *Ant*, GRD obtains the highest average accumulative reward of $1.5 \times 10^4$ and $6.5 \times 10^3$, separately, while the baseline methods achieve the best value of $1.25 \times 10^4$ and $6 \times 10^3$, respectively. In the task where the agents are equipped with high-dimension states, such as *HumanoidStandup* of 376-dimension states and *Ant* of

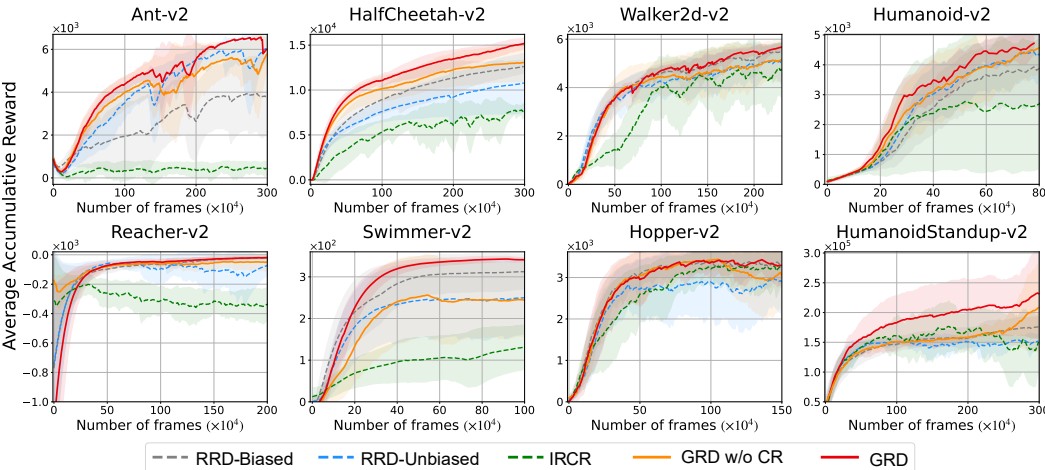

Figure 3: Learning curves on a suite of MuJoCo benchmark tasks with episodic rewards, based on 5 independent runs with random initialization. The shaded region indicates the standard deviation and the curves were smoothed by averaging the 10 most recent evaluation points using an exponential moving average. An evaluation point was established every $10^4$ time steps.

Table 1: The mean and the standard variance of average accumulative reward and sparsity rate $S$ regarding diverse $\lambda_1$ at different time step $t$ in *Swimmer*.

| $\lambda_1 \ / \ t$ | 1e5 | 2e5 | 3e5 | 5e5 | 8e5 | 1e6 |
|---|---|---|---|---|---|---|
| 0 | 87 ± 41(0.77) | 153± 43(0.80) | 151± 32(0.80) | 131± 55(0.80) | 170± 36(0.77) | 159± 25(0.78) |
| 3e-6 | **182±103(0.72)** | 201±109(0.71) | 217±108(0.72) | 217±100(0.68) | 229± 95(0.52) | 220±102(0.50) |
| 5e-6 | 130±103(0.56) | **245±104(0.69)** | **261±100(0.74)** | **286± 77(0.77)** | **272± 78(0.75)** | **262± 94(0.66)** |
| 5e-5 | 118±100(0.46) | 109±107(0.41) | 123±103(0.34) | 141± 96(0.25) | 152± 93(0.19) | 158± 90(0.17) |

111-dimension states, GRD gains significant improvement. Taking *HumanoidStandup* as an example, GRD achieves a higher average accumulative reward of $2.3 \times 10^5$ at the end of the training and always demonstrates better performance at a certain time step during the training. The possible explanation is that GRD can quickly learn the grounded causal structure and filter out the nearly useless dimensions for approximating the real reward function. The visualization of the learned causal structure and the related discussion can be found in Sec. 6.4.

## 6.3 Ablation Study

In this section, we first demonstrate that the compact representation improves policy learning and then investigate the potential impact of varying causal structures on policy learning.

**Compact representation for policy learning.** According to Figure 3, although the GRD w/o CR has access to more information (it takes all the dimensions of the state as the input of policy model), GRD earns higher accumulative rewards on all the tasks. This is because only the dimensions tied to the predicted Markovian rewards reflect the agents' goals. That is, the agent is supervised by the learned reward function while choosing an action based on the associated state dimensions, ensuring the input of the policy model aligns with its supervision signals. This constrains the policy model to be optimized over a smaller state space, leading to a more succinct and efficient training process.

**The impact of learned causal structure.** Define sparsity of causal structure of the state to reward, $S = \frac{1}{|\boldsymbol{s}|} \sum_{i=1}^{|\boldsymbol{s}|} \boldsymbol{C}_i^{\boldsymbol{s} \to r}$. We control the value of $\lambda_1$ to obtain different sparsities of learned causal structure. The average accumulative reward and the sparsity of causal structure during the training process in *Swimmer* are presented in Table 1. With the increase of $\lambda_1$ (like from $5e-6$ to $5e-5$), the causal structure generally gets more sparse (the sparsity $S$ decreases), leading to less policy improvement. The most possible explanation is, GRD does not take enough states to predict Markovian rewards and thus can not reflect the true goal of the agent, leading to misguided policy learning. By contrast, if we set a relatively low $\lambda_1$ to form a denser structure, GRD may consider redundant dimensions that harm policy learning. Therefore, a reasonable causal structure for the reward function can improve both the convergence speed and the performance of policy training.

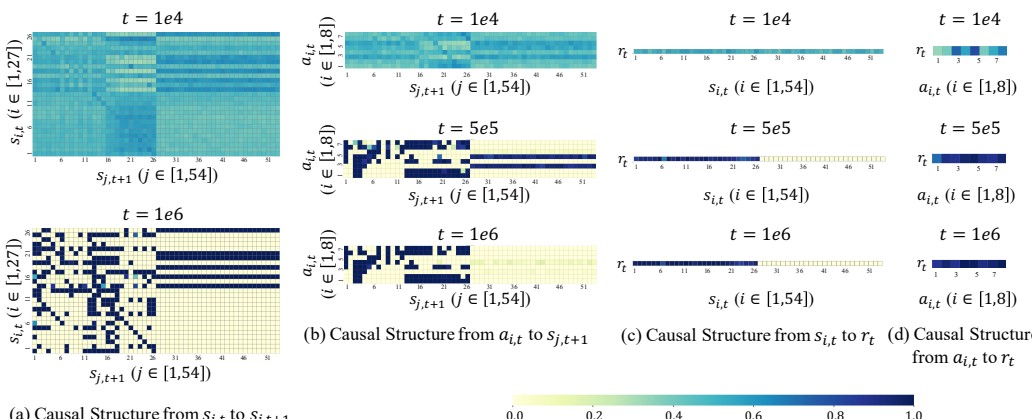

(a) Causal Structure from $s_{i,t}$ to $s_{j,t+1}$

(b) Causal Structure from $a_{i,t}$ to $s_{j,t+1}$

(c) Causal Structure from $s_{i,t}$ to $r_t$

(d) Causal Structure from $a_{i,t}$ to $r_t$

Figure 4: The visualization of learned causal structure for *Ant* when $t \in [1e4, 5e5, 1e6]$. The color indicates the probability of the existence of causal edges, whereas darker colors represent higher probabilities. There are 111 dimensions in the state variables, but only the first 27 ones are used. (a) The learned causal structure among the first 27 dimensions of the state variable $s_t$ to the first 54 dimensions of the next state variable $s_{t+1}$. Due to the limited space, we only visualize the structure at $t = 1e4$ and $t = 1e6$. (b) The learned causal structure among all dimensions of the action variable $a_t$ to the first 54 dimensions of the next state variable $s_{t+1}$. (c) The learned causal structure among the first 54 dimensions of the state variable $s_t$ to the Markovian reward variable $r_t$. (d) The learned causal structure among all dimensions of action variable $a_t$ to the Markovian reward variable $r_t$.

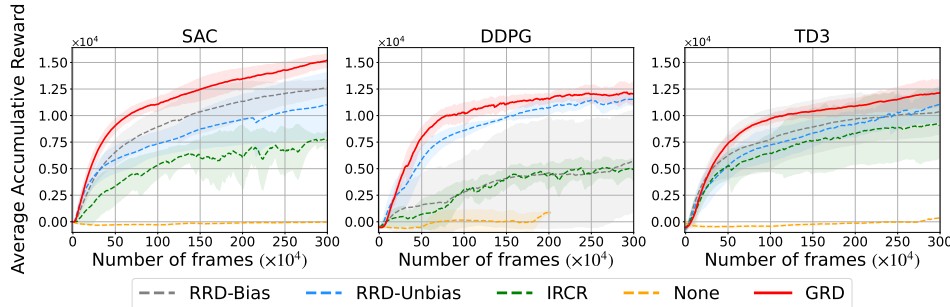

Figure 5: Evaluation with different RL backbones, SAC, DDPG and TD3. "None" is training with the observed delayed rewards.

**Robustness.** To evaluate the robustness of GRD, we provide results under noisy states in *Ant*. A significant characteristic of *Ant* is that only the first 28 dimensions of state are used. Therefore, the noise on the other dimensions ($28 \sim 111$) should not impact a learned robust policy. To verify this, during policy evaluation, we introduce the independent Gaussian noises with the mean of 0 and standard deviation of $0 \sim 1$ into those insignificant dimensions ($28 \sim 111$). As shown in Figure 6, GRD is unaffected by the injected noises, while the performance of the baseline methods decreases. That is because the insignificant dimensions are not in the compact representation, which serves as the policy input of GRD. Therefore, our method learns a more robust policy with the usage of compact representation.

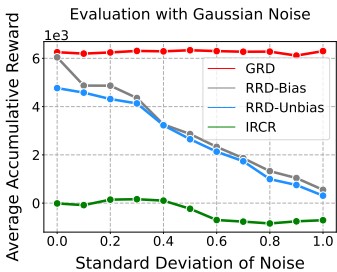

Figure 6: Evaluation with Gaussian Noise in the State.

**Results over different RL backbones.** We provide the results of training with DDPG [56] and TD3 [57] in Figure 5. As the experimental result shows, on the tasks of *HalfCheetah*, GRD consistently outperforms the baseline methods, RRD-Bias, RRD-Unbias, and IRCR, which are modified to run based on the same policy optimization algorithm, DDPG and TD3. We also provide results of "None", which utilizes the observed delayed reward for policy learning directly.

**Consistent improvement.** We provide results to showcase the consistent improvement of our method. 1) On different RL backbones: we provide results to demonstrate the consistent improvement of our method over different RL backbones (DDPG [56] and TD3 [57]) as shown in Figure 5; 2) On manipulation tasks: we provide the results on manipulation tasks in MetaWorld [58]. Please refer to Appendix D.

### 6.4 Visualization

We visualize the learned causal structure and redistributed rewards in *Ant*, where the robot observes 111-dimension states and 84 dimensions are not used [55]. The actions are 8-dimensional.

**Causal Structure.** As Figure 4, the existence of each element in the causal structure becomes more deterministic (indicated by the more distinguishable color) and further filters out more elements with the training. At the $1e4$ time step, GRD regards 75% elements in $\boldsymbol{C^{s \to r}} \in \{0,1\}^{376}$, 70% dimensions in $\boldsymbol{C^{a \to s}} \in$

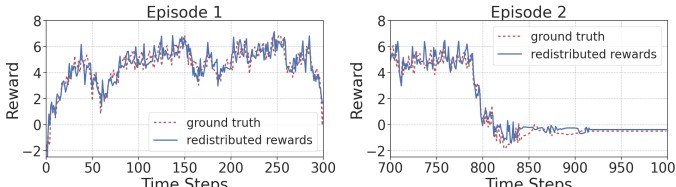

Figure 7: The visualization of decomposed rewards (blue solid lines) and the grounded rewards (red dotted lines).

$\{0,1\}^{8 \times 376}$, 98% dimensions in $\boldsymbol{C^{s \to s}} \in \{0,1\}^{376 \times 376}$ to be 0 with high probabilities, indicating that the causal edge not exists. The learned structure is nearly consistent with the fact that there should not be an edge from the unused dimensions of the state variable to the other variables. Although some redundant edges have been learned here (as shown in Figure 4 (a), the grids for the causal structure from some dimensions of state to the $28 \sim 54$ dimensions of the next state are of dark color), these edges do not affect our policy learning, since there is no edge from the $28 \sim 54$ dimensions of $\boldsymbol{s}_t$ to $r_t$, *i.e.*, these dimensions do not exist in the identified compact representation. Additionally, according to Figure 4(d), the edges from different dimensions of $\boldsymbol{a}$ to $r$ always exist. It corresponds to the reward design in *Ant* since the robots are expected to act with the lowest control cost, defined as $r_{t,cost} = \sum_{i=1}^{|\boldsymbol{a}|} \boldsymbol{a}_i^2$. Therefore, the learned causal structure captures some causal characteristics of the environment, which can explain the generation of the Markovian rewards.

**Decomposed Rewards.** We visualize the decomposed rewards and the ground truth rewards to demonstrate the accuracy of predicting Markovian reward by GRD. According to Figure 7, the redistributed rewards (blue lines) consistently align with the ground truth (red lines), indicating that our method indeed distinguishes the state-action pairs with less contribution to the long-term return from those with more contributions. More cases are provided in Appendix D.2.

## 7 Conclusion

In this paper, we propose Generative Return Decomposition (GRD) to address reward redistribution for the RL tasks with delayed rewards in an interpretable and principled way. GRD reformulates reward redistribution from a causal view and recovers the generative process within the MDPs with trajectory-wise return, supported by theoretical evidence. Furthermore, GRD forms a compact representation for efficient policy learning. Experimental results show that, in GRD, not only does the explainable reward redistribution produce a qualified supervision signal for policy learning, but also the identified compact representation promotes policy optimization.

**Limitation.** The limitations of our work arise from the made assumption. We presumed a stationary reward function, limiting our method's use in the case of a dynamic, non-stationary reward function. The assumed static causal graph and used causal discovery methods might not cater to situations with confounders. Besides, an important direction for future work is to extend our proposed method in the case of partial observable MDP and to discuss the identifiability in such scenarios.

**Broader Impacts.** Our motivation is to strive to make decisions that are both understood and trusted by humans. By enhancing the transparency and credibility of algorithms, we aim to harmonize human-AI collaboration to enable more reliable and responsible decision-making in various fields. Our GRD algorithm notably advances this by offering a causal view of reward generation and identifying a compact representation for policy learning.

## Acknowledgments

We thank the reviewers for the constructive comments and questions, which improved the quality of our paper. Part of this work used the Dutch national e-infrastructure with the support of the SURF Cooperative using grant no. EINF-4550. Yali Du thanks to the support by the EPSRC grant EP/Y003187/1.

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

# A  Background on Causal Inference

A directed acyclic graph (DAG), $\mathcal{G} = (\boldsymbol{V}, \boldsymbol{E})$, can be deployed to represent a graphical criterion carrying out a set of conditions on the paths, where $\boldsymbol{V}$ and $\boldsymbol{E}$ denote the set of nodes and the set of directed edges, separately.

**Definition 1 (d-separation [59])** *A set of nodes $\boldsymbol{Z} \subseteq \boldsymbol{V}$ blocks the path $p$ if and only if (1) $p$ contains a chain $i \to m \to j$ or a fork $i \leftarrow m \to j$ such that the middle node $m$ is in $\boldsymbol{Z}$, or (2) $p$ contains a collider $i \to m \leftarrow j$ such that the middle node $m$ is not in $\boldsymbol{Z}$ and such that no descendant of $m$ is in $\boldsymbol{Z}$. Let $\boldsymbol{X}$, $\boldsymbol{Y}$ and $\boldsymbol{Z}$ be disjunct sets of nodes. If and only if the set $\boldsymbol{Z}$ blocks all paths from one node in $\boldsymbol{X}$ to one node in $\boldsymbol{Y}$, $\boldsymbol{Z}$ is considered to d-separate $\boldsymbol{X}$ from $\boldsymbol{Y}$, denoting as $(\boldsymbol{X} \perp\!\!\!\perp_d \boldsymbol{Y} \mid \boldsymbol{Z})$.*

**Definition 2 (Global Markov Condition [59, 60])** *If, for any partition $(\boldsymbol{X}, \boldsymbol{Y}, \boldsymbol{Z})$, $\boldsymbol{X}$ is d-separated from $\boldsymbol{Y}$ given $\boldsymbol{Z}$, i.e., $\boldsymbol{X} \perp\!\!\!\perp_d \boldsymbol{Y} \mid \boldsymbol{Z}$. Then the distribution $P$ over $\boldsymbol{V}$ satisfies the global Markov condition on graph $G$, and can be factorizes as, $P(\boldsymbol{X}, \boldsymbol{Y} \mid \boldsymbol{Z}) = P(\boldsymbol{X} \mid \boldsymbol{Z})P(\boldsymbol{Y} \mid \boldsymbol{Z})$. That is, $\boldsymbol{X}$ is conditionally independent of $\boldsymbol{Y}$ given $\boldsymbol{Z}$, writing as $\boldsymbol{X} \perp\!\!\!\perp \boldsymbol{Y} \mid \boldsymbol{Z}$.*

**Definition 3 (Faithfulness Assumption [59, 60])** *The variables, which are not entailed by the Markov Condition, are not independent of each other.*

*Under the above assumptions, we can apply d-separation as a criterion to understand the conditional independencies from a given DAG $\mathcal{G}$. That is, for any disjoint subset of nodes $\boldsymbol{X}, \boldsymbol{Y}, \boldsymbol{Z} \subseteq \boldsymbol{V}$, $(\boldsymbol{X} \perp\!\!\!\perp \boldsymbol{Y} \mid \boldsymbol{Z})$ and $\boldsymbol{X} \perp\!\!\!\perp_d \boldsymbol{Y} \mid \boldsymbol{Z}$ are the necessary and sufficient condition of each other.*

# B  Details of Theoretical Analysis

**Proposition 1 (Identifiability)** *Suppose the state $\boldsymbol{s}_t$, action $\boldsymbol{a}_t$, trajectory-wise long-term return $R$ are observable while Markovian rewards $r_t$ are unobservable, and they form an MDP, as described in Eq. 2. Then under the global Markov condition and faithfulness assumption, the reward function $g$ and the Markovian rewards $r_t$ are identifiable, as well as the causal structure that is characterized by binary masks $\boldsymbol{C}^{\cdot \to \cdot}$ and $\boldsymbol{C}^{\cdot \to \cdot}$ and the transition dynamics $f$.*

Below is the proof of Proposition 1. We begin by clarifying the assumptions we made and then provide the mathematical proof.

**Assumption** We assume that, $\epsilon_{s,i,t}$ and $\epsilon_{r,t}$ in Eq. 2 are i.i.d additive noise. From the weight-space view of Gaussian Process [61], equivalently, the causal models for $\boldsymbol{s}_{i,t+1}$ and $r_t$ can be represented as follows, respectively,

$$\boldsymbol{s}_{i,t+1} = f_i(\boldsymbol{s}_t, \boldsymbol{a}_t) + \epsilon_{s,i,t} = W_{i,f}^T \phi_f(\boldsymbol{s}_t, \boldsymbol{a}_t) + \epsilon_{s,i,t}, \tag{9}$$

$$r_t = g(\boldsymbol{s}_t, \boldsymbol{a}_t) + \epsilon_{r,t} = W_g^T \phi_g(\boldsymbol{s}_t, \boldsymbol{a}_t) + \epsilon_{r,t}, \tag{10}$$

where $\forall i \in [1, |\boldsymbol{s}|]$, and $\phi_f$ and $\phi_g$ denote basis function sets.

Then we denote the variable set in the system by $\boldsymbol{V}$, with $\boldsymbol{V} = \{\boldsymbol{s}_{1,t}, \ldots, \boldsymbol{s}_{|\boldsymbol{s}|,t}, \boldsymbol{a}_{1,t}, \ldots, \boldsymbol{a}_{|\boldsymbol{a}|,t}, r_t\}_{t=1}^T \cup R$, and the variables form a Bayesian network $\mathcal{G}$. Note, we assume that there are possible edges only from $\boldsymbol{s}_{i,t-1} \in \boldsymbol{s}_{t-1}$ to $\boldsymbol{s}_{i',t} \in \boldsymbol{s}_t$, from $\boldsymbol{a}_{j,t-1} \in \boldsymbol{a}_{t-1}$ to $\boldsymbol{s}_{i',t} \in \boldsymbol{s}_t$, from $\boldsymbol{s}_{i,t} \in \boldsymbol{s}_t$ to $r_t$, and from $\boldsymbol{a}_{j,t} \in \boldsymbol{a}_t$ to $r_t$ in $\mathcal{G}$. In particular, the $r_t$ are unobserved, while $R = \sum_{t=1}^T \gamma^{t-1} o_t$ is observed. Thus, there are deterministic edges from each $r_t$ to $R$.

Below we omit the $\gamma$ for simplicity.

**Proof 1** *Given trajectory-wise long-term return $R$, the binary masks, $\boldsymbol{C}^{\boldsymbol{s} \to r}$, $\boldsymbol{C}^{\boldsymbol{a} \to r}$ and Markovian reward function $g$ and the rewards $r_t$ are identifiable.* Following the above assumption, we first rewrite the function to calculate trajectory-wise long-term return in Eq. 2 as,

$$
\begin{aligned}
R &= \sum_{t=1}^T r_t \\
&= \sum_{t=1}^T \left[ W_g^T \phi_g(\boldsymbol{s}_t, \boldsymbol{a}_t) + \epsilon_{r,t} \right] \\
&= W_g^T \sum_{t=1}^T \phi_g(\boldsymbol{s}_t, \boldsymbol{a}_t) + \sum_{t=1}^T \epsilon_{r,t}.
\end{aligned} \tag{11}
$$

*For simplicity, we replace the components in Eq. 11 by,*

$$
\begin{aligned}
\zeta_g(X) &= \sum_{t=1}^{T} \phi_g(\boldsymbol{s}_t, \boldsymbol{a}_t), \\
E_r &= \sum_{t=1}^{T} \epsilon_{r,t},
\end{aligned}
\tag{12}
$$

*where $X := [\boldsymbol{s}_t, \boldsymbol{a}_t]_{t=1}^{T}$ representing the concatenation of the covariates $\boldsymbol{s}_t$ and $\boldsymbol{a}_t$ from $t = 1$ to $T$. Consequently, we derive the following equation,*

$$
R = W_g^T \zeta_g(X) + E_r.
\tag{13}
$$

*Then we can obtain a closed-form solution of $W_g^T$ in Eq. 13 by modeling the dependencies between the covariates $X_\tau$ and response variables $R_\tau$, where both are continuous. One classical approach to finding such a solution involves minimizing the quadratic cost and incorporating a weight-decay regularizer to prevent overfitting. Specifically, we define the cost function as,*

$$
C(W_g) = \frac{1}{2} \sum_{X_\tau, R_\tau \sim \mathcal{D}} (R_\tau - W_g^T \zeta_g(X_\tau))^2 + \frac{1}{2} \lambda \|W_g\|^2.
\tag{14}
$$

*where $\tau$ represents trajectories consisting of state-action pairs $X_\tau$ and long-term returns $R_\tau$, which are sampled from the replay buffer $\mathcal{D}$. $\lambda$ is the weight-decay regularization parameter. To find the closed-form solution, we differentiate the cost function with respect to $W_g$ and set the derivative to zero:*

$$
\frac{\partial C(W_g)}{\partial W_g} = 0.
\tag{15}
$$

*Solving this equation will yield the closed-form solution for $W_g^T$, i.e.,*

$$
W_g = (\lambda I_d + \zeta_g \zeta_g^T)^{-1} \zeta_g R = \zeta_g (\zeta_g^T \zeta_g + \lambda I_n)^{-1} R
\tag{16}
$$

*Therefore, $W_g$, which indicates the causal structure and strength of the edge, can be identified from the observed data. In summary, given trajectory-wise long-term return $R$, the binary masks, $\boldsymbol{C}^{s \to r}$, $\boldsymbol{C}^{a \to r}$ and Markovian reward function $g$ and the rewards $r_t$ are identifiable.*

**The binary masks, $\boldsymbol{C}^{s \to s}$, $\boldsymbol{C}^{a \to s}$ and the transition dynamics $f$ are identifiable** *In a similar manner, based on the assumption and Eq. 2, we can rewrite Eq. 9 to,*

$$
\boldsymbol{s}_{t+1} = W_{i,f}^T \phi_f(\boldsymbol{s}_t, \boldsymbol{a}_t) + \epsilon_{s,i,t}.
\tag{17}
$$

*To obtain a closed-form solution for $W{i, f}^T$ in Equation 17, we can model the dependencies between the covariates $X_t$ and the response variables $\boldsymbol{s}_{t+1}$, both of which are continuous. The closed-form solution can be represented as:*

$$
C(W_{i,f}) = \frac{1}{2} \sum_{\boldsymbol{s}_{i,t}, \boldsymbol{s}_{i,t+1} \sim \mathcal{D}} (\boldsymbol{s}_{i,t+1} - W_{i,f}^T \phi_{i,f}(\boldsymbol{s}_t, \boldsymbol{a}_t))^2 + \frac{1}{2} \lambda \|W_{i,f}\|^2.
\tag{18}
$$

*By taking derivatives of the cost function and setting them to zero, we can obtain the closed-form solution,*

$$
\begin{aligned}
W_{i,f} &= (\lambda I_d + \phi_{i,f} \phi_{i,f}^T)^{-1} \phi_{i,f} \boldsymbol{s}_{i,t+1} \\
&= \phi_{i,f} (\phi_{i,f}^T \phi_{i,f} + \lambda I_n)^{-1} \boldsymbol{s}_{i,t+1}.
\end{aligned}
\tag{19}
$$

*Therefore, $W_{i,f}$ can be identified from the observed data. This conclusion applies to all dimensions of the state. As a result, the $f$, which indicates the parent nodes of the $i$-dimension of the state, as well as the strength of the causal edge, are identifiable. In summary, the binary masks, $\boldsymbol{C}^{s \to s}$, $\boldsymbol{C}^{a \to s}$ and the transition dynamics $f$ are identifiable.*

*Considering the Markov condition and faithfulness assumption, we can conclude that for any pair of variables $V_i, V_j \in \boldsymbol{V}$, $V_i$ and $V_j$ are not adjacent in the causal graph $\mathcal{G}$ if and only if they are conditionally independent given some subset of $\{V_l \mid l \neq i, l \neq j\}$. Additionally, since there are no instantaneous causal relationships and the direction of causality can be determined if an edge exists, the binary structural masks $\boldsymbol{C}^{s \to r}$, $\boldsymbol{C}^{a \to r}$, $\boldsymbol{C}^{s \to s}$, and $\boldsymbol{C}^{a \to s}$ defined over the set $\boldsymbol{V}$ are identifiable with conditional independence relationships [30]. Consequently, the functions $f$ and $g$ in Equation 2 are also identifiable.*

**Algorithm 1** Learning the generative process and policy jointly.

---

1: Initialize: Environment $\mathcal{E}$, trajectory $\tau \leftarrow \emptyset$, buffer $\mathcal{D} \leftarrow \emptyset$
2: Initialize: Generative Model $\Phi_{\mathrm{m}} := [\phi_{\mathrm{cau}}, \phi_{\mathrm{dyn}}, \phi_{\mathrm{rew}}]$; Policy Model $\Phi_{\pi}$
3: **for** $i = 1, 2, \ldots, 3 \times 10^4$ **do**
4:     $\tau \leftarrow \emptyset$, reset $\mathcal{E}$
5:     **for** $n_{\mathrm{step}} = 1, 2, \ldots, 100$ **do**
6:         Sample data $\langle \boldsymbol{s}_t, \boldsymbol{a}_t, o_t \rangle$ from $\mathcal{E}$, and store them to trajectory $\tau$
7:
8:         **if** $\mathcal{E}$ done **then**
9:             store trajectory $\tau = \{\boldsymbol{s}_{1:T}, \boldsymbol{a}_{1:T}, R\}$ to buffer $\mathcal{D}$, where $R = \sum_{i=1}^{T} \gamma^{t-1} o_t$
10:             $\tau \leftarrow \emptyset$, reset $\mathcal{E}$
11:         **end if**
12:
13:         **for** $n_{\mathrm{batch}} = 1, 2, \ldots, \mathrm{train\_batches}$ **do**
14:
15:             // Optimize Generative Model $\Phi_{\mathrm{m}}$
16:             Sample $D_1$ consisting of $M$ trajectories from $\mathcal{D}$: $D_1 = \{\langle \boldsymbol{s}_t^m, \boldsymbol{a}_t^m \rangle \mid_{t=1}^T, R^m\} \mid_{m=1}^M$
17:             Sample binary masks by Gumbel-Softmax from $\phi_{\mathrm{cau}}$: $\boldsymbol{C}^{\boldsymbol{s} \to r}$ and $\boldsymbol{C}^{\boldsymbol{a} \to r}$
18:             Optimize $\phi_{\mathrm{rew}}$ with $D_1$: $\phi_{\mathrm{rew}} \leftarrow \phi_{rew} - \alpha \nabla_{\phi_{\mathrm{rew}}} L_{\mathrm{rew}}$ (Eq. 4)
19:
20:             Sample $D_2$ consisting of $N$ samples from $\mathcal{D}$: $D_2 = \{\boldsymbol{s}_{t_n}, \boldsymbol{a}_{t_n}, \boldsymbol{s}_{t_n+1}\} \mid_{n=1}^N$
21:             Sample binary masks by Gumbel-Softmax from $\phi_{\mathrm{cau}}$: $\boldsymbol{C}^{\boldsymbol{s} \to r}, \boldsymbol{C}^{\boldsymbol{a} \to r}, \boldsymbol{C}^{\boldsymbol{s} \to \boldsymbol{s}}$ and $\hat{\boldsymbol{C}}^{\boldsymbol{a} \to \boldsymbol{s}}$
22:             Optimize $\phi_{\mathrm{dyn}}$ with $D_2$ (Using $\boldsymbol{C}^{\boldsymbol{s} \to \boldsymbol{s}}$ and $\boldsymbol{C}^{\boldsymbol{s} \to r}$): $\phi_{\mathrm{dyn}} \leftarrow \phi_{dyn} - \alpha \nabla_{\phi_{dyn}} L_{dyn}$ (Eq. 5)
23:
24:             Optimize $\phi_{\mathrm{cau}}$: $\phi_{\mathrm{cau}} \leftarrow \phi_{\mathrm{cau}} - \alpha \nabla_{\phi_{\mathrm{cau}}} (L_{\mathrm{sp}} + L_{\mathrm{rew}} + L_{\mathrm{dyn}})$ (Eq. 6)
25:
26:             // Optimize Policy Model $\Phi_{\pi}$
27:             Sample binary masks greedily from $\phi_{\mathrm{cau}}$: $\boldsymbol{C}^{\boldsymbol{s} \to r}, \boldsymbol{C}^{\boldsymbol{a} \to r}$, and $\boldsymbol{C}^{\boldsymbol{s} \to \boldsymbol{s}}$
28:             Calculate $\boldsymbol{C}^{\boldsymbol{s} \to \pi}$ based on $\boldsymbol{C}^{\boldsymbol{s} \to r}$ and $\boldsymbol{C}^{\boldsymbol{s} \to \boldsymbol{s}}$
29:             Update $D_2$: $D_2 \leftarrow \{\boldsymbol{C}^{\boldsymbol{s} \to \pi} \odot \boldsymbol{s}_{t_n}, \boldsymbol{a}_{t_n}, \boldsymbol{s}_{t_n+1}, \phi_{\mathrm{rew}}(\boldsymbol{s}_{t_n}, \boldsymbol{a}_{t_n}, \boldsymbol{C}^{\boldsymbol{s} \to r}, \boldsymbol{C}^{\boldsymbol{a} \to r})\} \mid_{n=1}^N$
30:             Optimize $\Phi_{\pi}$: $\Phi_{\pi} \leftarrow \Phi_{\pi} - \alpha \nabla_{\Phi_{\pi}} J_{\pi}$ (Eq. 8)
31:
32:         **end for**
33:     **end for**
34: **end for**

---

# C   Details of Implementation

## C.1   Baselines

We compare our method against the following baselines,

- RRD (biased). This baseline utilizes a surrogate objective called randomized return decomposition loss for reducing the consumption of estimating the Markovian reward function. It applies Monte-Carlo sampling to get a biased estimation of the Mean Square Error (MSE) between the observed episodic reward and the sum of Markovian reward predictions in a sequence. We keep the same setting and hyper-parameters with its official implementation to reproduce the results, in which the policy module is optimized by soft actor-critic (SAC) algorithm [50].

- RRD (unbiased). This variant of RRD (biased) provides an unbiased estimation of MSE by sampling short sub-sequences. It offers a computationally efficient approach to optimize MSE. According to [17], RRD (biased) and RRD (unbiased) achieve state-of-the-art performance in episodic MuJoCo tasks.

- This baseline performs non-parametric uniform reward redistribution. At each time step, the proxy reward is set to the normalized value of the trajectory return. IRCR is a simple and efficient approach, and except for RRD, it achieves state-of-the-art performance in the literature. The implementation is from RRD [17].

## C.2   Detailed Generative Model

The parametric generative model $\Phi_{\mathrm{m}}$ used in the MDP environment consists of three components: $\phi_{\mathrm{cau}}$, $\phi_{\mathrm{rew}}$, and $\phi_{\mathrm{dyn}}$. We provide a detailed description of their model structures below.

| Layer# | 1 | 2 | 3 |
|---|---|---|---|
| $\phi_{\text{rew}}$ | FC256 | FC256 | FC1 |
| $\phi_{\text{dyn}}$ | FC256 | FC256 | FC9 |
| $\phi_{\pi}$ | FC256 | FC256 | FC2$|\boldsymbol{a}|$ |
| $\phi_{v}$ | FC256 | FC256 | FC1 |

Table 2: The network structures of $\phi_{\text{rew}}$, $\phi_{\text{dyn}}$, $\phi_{\pi}$ and $\phi_{v}$. $FC256$ denotes a fully-connected layer with an output size of 256. Each hidden layer is followed by an activation function, ReLU. $|\boldsymbol{a}|$ is the number of dimensions of the action in a specific task.

$\phi_{\textbf{cau}}$ **for predicting the causal structure.** $\phi_{\text{cau}}$ comprises a set of free parameters without input. We divide $\phi_{\text{cau}}$ into four parts, each corresponding to the binary masks in Equation 2. Specifically, we have

- $\phi_{\text{cau}}^{\boldsymbol{s}\to\boldsymbol{s}} \in \mathbb{R}^{|\boldsymbol{s}|\times|\boldsymbol{s}|\times 2}$ for $\boldsymbol{C}^{\boldsymbol{s}\to\boldsymbol{s}} \in \{0,1\}^{|\boldsymbol{s}|\times|\boldsymbol{s}|}$,
- $\phi_{\text{cau}}^{\boldsymbol{a}\to\boldsymbol{s}} \in \mathbb{R}^{|\boldsymbol{a}|\times|\boldsymbol{s}|\times 2}$ for $\boldsymbol{C}^{\boldsymbol{a}\to\boldsymbol{s}} \in \mathbb{R}^{|\boldsymbol{a}|\times|\boldsymbol{s}|}$,
- $\phi_{\text{cau}}^{\boldsymbol{s}\to r} \in \mathbb{R}^{|\boldsymbol{s}|\times 2}$ for $\boldsymbol{C}^{\boldsymbol{s}\to r} \in \mathbb{R}^{|\boldsymbol{s}|}$,
- $\phi_{\text{cau}}^{\boldsymbol{a}\to r} \in \mathbb{R}^{|\boldsymbol{a}|\times 2}$ for $\boldsymbol{C}^{\boldsymbol{a}\to r} \in \mathbb{R}^{|\boldsymbol{a}|}$.

Below we explain the shared workflows in $\phi_{\text{cau}}$ using the example of predicting the causal edge from the $i$-th dimension of state $\boldsymbol{s}_{i,t}$ to the $j$-th dimension of the next state $\boldsymbol{s}_{j,t+1}$, by part of the free parameters, $\phi_{\text{cau},i,j}^{\boldsymbol{s}\to\boldsymbol{s}}$.

For simplicity, we denote $\phi_{\text{cau},i,j}^{\boldsymbol{s}\to\boldsymbol{s}}$ as $\psi$. The shape of $\psi$ is now easy to be determined. That is $\psi \in \mathbb{R}^2$ and we write it as $\psi = [\psi_0, \psi_1]$. With this 2-element vector, we can characterize a Bernoulli distribution, where each element corresponds to the unnormalized probability of classifying the edge as existing ($\psi_0$) or not existing ($\psi_1$), respectively. Therefore, the probability of the causal edge existing from the $i$-th dimension of state $\boldsymbol{s}_{i,t}$ to the $j$-th dimension of the next state $\boldsymbol{s}_{j,t+1}$ can be calculated as:

$$P(\boldsymbol{C}_{i,j}^{\boldsymbol{s}\to\boldsymbol{s}}) = \frac{\exp(\psi_0)}{\exp(\psi_0) + \exp(\psi_1)}. \tag{20}$$

One example of using the estimated mask is given as Figure 8.

During training: obtain $\boldsymbol{C}_{i,j}^{\boldsymbol{s}\to\boldsymbol{s}}$ through Gumbel-Softmax sampling. In the training phase, it is crucial to maintain the gradient flow for backpropagation. To achieve this, we sample the binary values of $\boldsymbol{C}_{i,j}^{\boldsymbol{s}\to\boldsymbol{s}}$ by applying Gumbel-Softmax [48],

$$\boldsymbol{C}_{i,j}^{\boldsymbol{s}\to\boldsymbol{s}} = \text{GS}(\psi) \tag{21}$$

where GS denotes the Gumbel-Softmax sampling, which allows us to obtain binary discrete samples from the Bernoulli distribution. Applying Gumbel Softmax sampling allows us to randomly sample from the Bernoulli distribution in a stochastic manner rather than simply selecting the class with the highest probability. This introduces some randomness, enabling the model to explore the balance and uncertainty between different classifications more flexibly.

The above explanation of the workflow in $\phi_{\text{cau}}$ for predicting a single causal edge provides insight into the overall implementation of the entire module $\phi_{\text{cau}}$ and can be applicable for predicting all the causal edges during the training phase. Therefore, we can obtain $\boldsymbol{C}^{\boldsymbol{a}\to\boldsymbol{s}}$ for optimizing $\phi_{\text{dyn}}$, $\boldsymbol{C}^{\boldsymbol{s}\to r}$ and $\boldsymbol{C}^{\boldsymbol{a}\to r}$ for optimizing $\phi_{\text{rew}}$, using similar procedures.

During inference: obtain $\boldsymbol{C}_{i,j}^{\boldsymbol{s}\to\boldsymbol{s}}$ by greedy selection. In the inference phase, including data sampling and policy learning, we get the prediction of $\boldsymbol{C}_{i,j}^{\boldsymbol{s}\to\boldsymbol{s}}$ through a greedy selection,

$$\boldsymbol{C}_{i,j}^{\boldsymbol{s}\to\boldsymbol{s}} = \begin{cases} 1, \psi_0 \geq \psi_1 \\ 0, \psi_0 < \psi_1. \end{cases} \tag{22}$$

Such a greedy sampling avoids introducing randomness using the Gumble-Softmax sampling.

The above explanation of the workflow in $\phi_{\text{cau}}$ for predicting a single causal edge provides insight into the overall implementation of the entire module $\phi_{\text{cau}}$ and can be applicable for predicting all the causal edges during the inference phase. Therefore, we can obtain $\boldsymbol{C}^{\boldsymbol{s}\to\boldsymbol{s}}$, $\boldsymbol{C}^{\boldsymbol{s}\to r}$ and $\boldsymbol{C}^{\boldsymbol{a}\to r}$ using similar procedures to predict the Markovian reward and extract compact representation $\boldsymbol{s}^{\min}$.

$\phi_{\textbf{rew}}$ **for predicting the Markovian rewards.** $\phi_{\text{rew}}$ is a stacked, fully-connected network, and the details of the network structure are provided in Table 2. During training, the prediction of Markovian reward can be

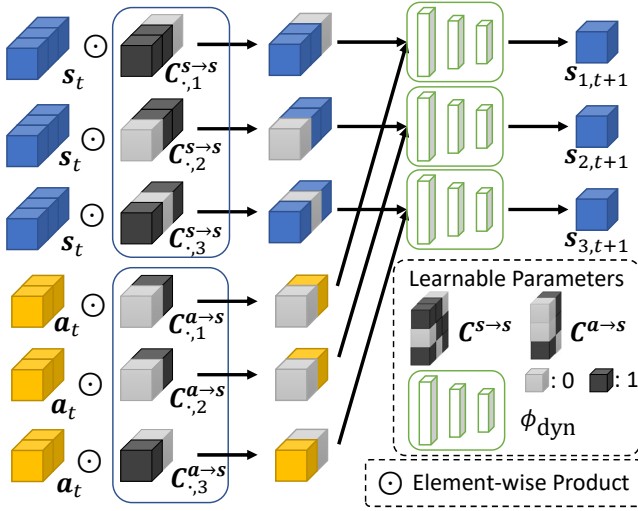

Figure 8: The illustration of using learnable masks to predict the next state.

written as,

$$\hat{r} = \phi_{\text{rew}}(\boldsymbol{s}_t, \boldsymbol{a}_t, \boldsymbol{C}^{\boldsymbol{s} \rightarrow r}, \boldsymbol{C}^{\boldsymbol{a} \rightarrow r}) = \text{FCs}([\boldsymbol{C}^{\boldsymbol{s} \rightarrow r} \odot \boldsymbol{s}_t, \boldsymbol{C}^{\boldsymbol{a} \rightarrow r} \odot \boldsymbol{a}_t]), \tag{23}$$

where $[\cdot, \cdot]$, $\odot$ denotes concatenation and element-wise multiply operations, respectively. FCs denotes the stacked fully-connected network. $\boldsymbol{C}^{\boldsymbol{s} \rightarrow r}$ and $\boldsymbol{C}^{\boldsymbol{s} \rightarrow r}$ are derived from $\phi_{\text{cau}}$ by Gumbel-Softmax.

During inference, including policy learning and data sampling, the predicted Markovian reward is

$$\hat{r} = \phi_{\text{rew}}(\boldsymbol{s}_t, \boldsymbol{a})_t, \boldsymbol{C}^{\boldsymbol{s} \rightarrow r}, \boldsymbol{C}^{\boldsymbol{a} \rightarrow r}) = \text{FCs}([\boldsymbol{C}^{\boldsymbol{s} \rightarrow r} \odot \boldsymbol{s}_t, \boldsymbol{C}^{\boldsymbol{a} \rightarrow r} \odot \boldsymbol{a}_t]), \tag{24}$$

where $\boldsymbol{C}^{\boldsymbol{s} \rightarrow r}$ and $\boldsymbol{C}^{\boldsymbol{s} \rightarrow r}$ are derived from $\phi_{\text{cau}}$ greedily by deterministic sampling.

$\phi_{\textbf{dyn}}$ **for modeling the environment dynamics.** In our experiment, we do not directly utilize $\phi_{\text{dyn}}$ in policy learning. Instead, this module serves as a bridge to optimize $\phi_{\text{cau}}^{\boldsymbol{s} \rightarrow \boldsymbol{s}}$ and $\phi_{\text{cau}}^{\boldsymbol{a} \rightarrow \boldsymbol{s}}$. Subsequently, $\phi_{\text{cau}}^{\boldsymbol{s} \rightarrow \boldsymbol{s}}$ can be utilized in the calculation of $\boldsymbol{C}^{\boldsymbol{s} \rightarrow \pi}$.

During training, we initially sample $\boldsymbol{C}^{\boldsymbol{s} \rightarrow \boldsymbol{s}}$ and $\boldsymbol{C}^{\boldsymbol{a} \rightarrow \boldsymbol{s}}$ using Gumbel-Softmax. The prediction for the $i$-th dimension of the next state can be represented as follows,

$$\hat{\boldsymbol{s}}_{i,t} = \text{MDN}([\boldsymbol{C}_{\cdot,i}^{\boldsymbol{s} \rightarrow \boldsymbol{s}} \odot \boldsymbol{s}_t, \boldsymbol{C}_{\cdot,i}^{\boldsymbol{a} \rightarrow \boldsymbol{s}} \odot \boldsymbol{a}_t]), \tag{25}$$

where $[\cdot, \cdot]$ denotes concatenation and MDN denotes the Mixture Density Network, which outputs the means, variances, and probabilities for $N_{Gau}$ Gaussian cores. The parameters of MDN are shared across the predictions of different dimensions of the next state. We set $N_{cau} = 3$ in our experiments. More details about $\phi_{\text{dyn}}$ can be found in Table 2.

## C.3 Detailed Policy Model

Considering the specific requirements of the employed RL algorithm, Soft Actor-Critic (SAC), our Policy Model $\Phi_\pi$ comprises two components, the actor $\phi_\pi$ and the critic $\phi_v$. Detailed network structures for both components can be found in Table 2.

## C.4 Training Process

We follow the line of joint learning in [17], which avoids learning a return decomposition model in advance using data sampled by optimal or sub-optimal policies [16]. During each mini-batch training iteration, we sample two sets of data separately from the replay buffer $\mathcal{D}$:

- $D_1 = \{\langle \boldsymbol{s}_t^m, \boldsymbol{a}_t^m \rangle |_{t=1}^T, R^m \} |_{m=1}^M$ consists of $M$ trajectories. Provided with the trajectory-wise long-term returns $R^m |_{m=1}^M$, $D_1$ is utilized to optimize $\phi_{\text{cau}}^{\boldsymbol{s} \rightarrow r}$, $\phi_{\text{cau}}^{\boldsymbol{a} \rightarrow r}$ and $\phi_{\text{rew}}$, with $L_{\text{rew}}$.

- $D_2 = \{\boldsymbol{s}_{t_n}, \boldsymbol{a}_{t_n}, \boldsymbol{s}_{t_n+1} \} |_{n=1}^N$ consists of $N$ state-action pairs. $D_2$ are used for policy optimization and optimize the parts of causal structure, $\phi_{\text{cau}}^{\boldsymbol{s} \rightarrow \boldsymbol{s}}$ and $\phi_{\text{cau}}^{\boldsymbol{a} \rightarrow \boldsymbol{s}}$, $\phi_{\text{dyn}}$. With such a $D_2$, GRD breaks the temporal cues in the training data to learn the policy and dynamics function.

Please refer to Algorithm 1 for a detailed training process.

Table 3: The table of the hyper-parameters used in the experiments for GRD.

| Envs | $\lambda_1$ | $\lambda_2$ | $\lambda_3$ | $\lambda_4$ | $\lambda_5$ |
|---|---|---|---|---|---|
| *Ant* | $10^{-5}$ | $0$ | $10^{-7}$ | $10^{-8}$ | $10^{-8}$ |
| *HalfCheetah* | $10^{-5}$ | $10^{-5}$ | $10^{-5}$ | $10^{-6}$ | $10^{-5}$ |
| *Walker2d* | $10^{-5}$ | $10^{-5}$ | $10^{-6}$ | $10^{-6}$ | $10^{-7}$ |
| *Humanoid* | $10^{-5}$ | $10^{-8}$ | $10^{-5}$ | $10^{-7}$ | $10^{-8}$ |
| *Reacher* | $5 \times 10^{-7}$ | $10^{-8}$ | $10^{-8}$ | $10^{-8}$ | $10^{-8}$ |
| *Swimmer* | $10^{-7}$ | $10^{-9}$ | $10^{-9}$ | $0$ | $10^{-9}$ |
| *Hopper* | $10^{-6}$ | $10^{-6}$ | $10^{-6}$ | $10^{-7}$ | $10^{-6}$ |
| *HumanStandup* | $10^{-5}$ | $10^{-4}$ | $10^{-6}$ | $10^{-7}$ | $10^{-7}$ |

Table 4: The hyper-parameters.

| hyperparameters | value | hyperparameters | value |
|---|---|---|---|
| epochs | 3 | optimizer | Adam |
| cycles | 100 | learning rate | $3 \times 10^{-4}$ |
| iteration | 100 | $N$ | 256 |
| train_batches | 100 | $M$ | 4 |
| replay buffer size | $10^6$ | $\gamma$ | 1.00 |
| evaluation episodes | 10 | Polyak-averaging coefficient | 0.0005 |

## C.5 Hyper-Parameters

The network is trained from scratch using the Adam optimizer, without any pre-training. The initial learning rate for both model estimation and policy learning is set to $3 \times 10^{-4}$. The hyperparameters for policy learning are shared across all tasks, with a discount factor of $1.00$ and a Polyak-averaging coefficient of $5 \times 10^{-4}$. The target entropy is set to the negative value of the dimension of the robot action. To facilitate training, we utilize a replay buffer with a size of $1 \times 10^6$ time steps. The warmup size of the buffer for training is set to $1 \times 10^4$. The model is trained for 3 epochs, with each epoch consisting of 100 training cycles. In each cycle, we repeat the process of data collection and model training for 100 iterations. During each iteration, we collect data from 100 time steps of interaction with the MuJoCo simulation, which is then stored in the replay buffer. For training the $\phi_{\text{rew}}$, we sample 4 episodes, each containing $5 \times 10^3$ steps. As for policy learning and the optimization of $\phi_{\text{dyn}}$, we use data from 256 time steps. $\phi_{\text{cau}}$ is trained together with $\phi_{\text{rew}}$ and $\phi_{\text{dyn}}$. Validation is performed after every cycle, and the average metric is computed based on 10 test rollouts. The hyperparameters for learning the GRD model can be found in Table 3. All experiments were conducted on an HPC system equipped with 128 Intel Xeon processors operating at a clock speed of 2.2 GHz and 5 terabytes of memory.

# D Additional Results

## D.1 Results over manipulation tasks

We provide the comparison of our method with RRD in the three tasks of MetaWorld, as shown in Figure 9. The evaluation metric is the success rate. As shown in the results, compared with RRD, GRD achieves comparable or better performance.

- Door Lock: The agent must lock the door by rotating the lock clockwise.

- Push Wall: The agent is required to bypass a wall and push a puck to a goal. The puck and goal positions are randomized.

- Pick Place: The agent needs to pick and place a puck to a goal. The puck and goal positions are randomized.

## D.2 Visualization of Decomposed Rewards

As shown in Figure 10, we visualize the redistributed rewards in *Ant* by GRD, as well as the grounded rewards provided by the environment.

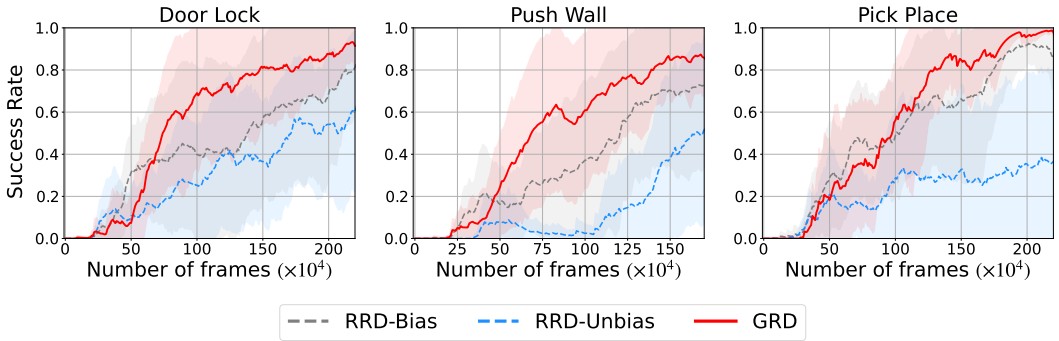

Figure 9: Evaluation on the manipulation tasks of MetaWorld, *Door Lock*, *Push Wall*, *Pick Place*.

### D.3 Visualization of Learned Causal Structure

We provide the visualization of learned causal structure in *Swimmer*, as shown in 11. Since the grounded causal structure is not accessible, we verify the reasonability of the learned causal structure by some observations:

- All the edges from different dimensions of $a$ to $r$ always exist, as shown in Figure 11 (d): *Swimmer* shares the same characteristic with *Ant* that the edges from different dimensions of $a$ to $r$ always exist, corresponding with the reward design of penalizing the agent if it takes actions that are too large, measured by the sum of the values of all the action dimensions.
- According to Figure 11 (b), the first dimension of action (Torque applied on the first rotor) has an impact on the last three dimensions of state (angular velocity of front tip, angular velocity of first rotor, second rotor), which is corresponding with the intuitive that the first dimension of action should impact the part that connects to first rotor. We can get a similar observation for the second action dimension from Figure 11.
- We can observe that all the state dimensions are learned to be connected to the reward; the possible explanation is that in the swimmer robot, any changes of the front tip, or two rotors will impact the position of the robot, potentially influencing the reward.

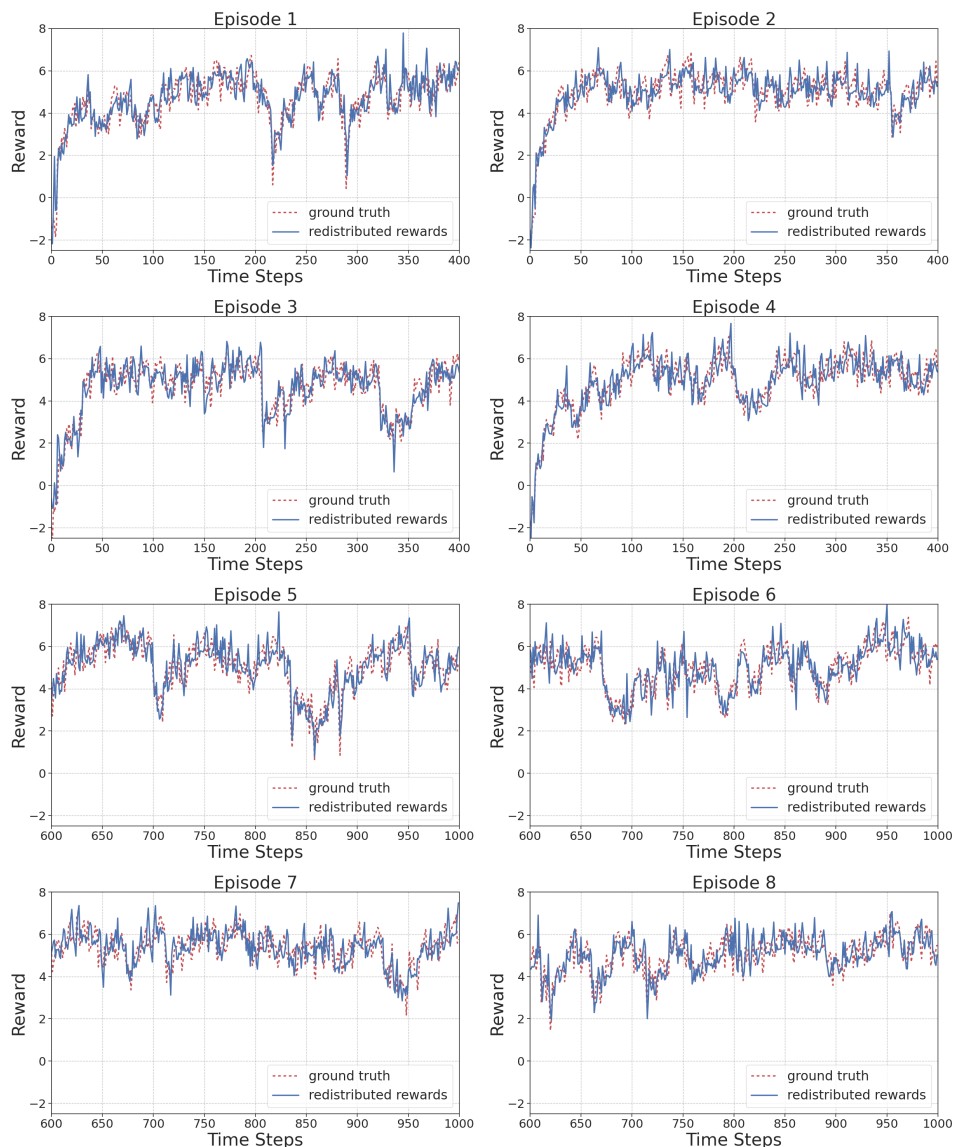

Figure 10: The visualization of redistributed rewards and grounded rewards in *Ant*. The results are produced by the GRD model trained after $1 \times 10^6$ steps. The redistributed rewards are shown in red, and the grounded rewards are shown in blue.

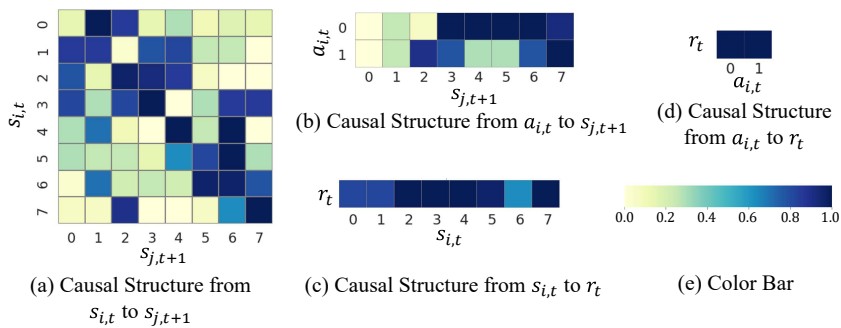

(a) Causal Structure from $s_{i,t}$ to $s_{j,t+1}$

(b) Causal Structure from $a_{i,t}$ to $s_{j,t+1}$

(c) Causal Structure from $s_{i,t}$ to $r_t$

(d) Causal Structure from $a_{i,t}$ to $r_t$

(e) Color Bar

Figure 11: Learned causal structure in *Swimmer-v2*.

