## A Broader Impacts

Our motivation is to strive to make decisions that are both understood and trusted by humans. By increasing the credibility and transparency of the decision-making process, human users develop a better understanding, validate and enhance the decisions made by algorithms. As a result, it is possible to bridge the gap between humans and AI, fostering a symbiotic relationship that leverages the strengths of both to enable more reliable and responsible decision-making in various filed. For example, in finance, transparent algorithms can enhance risk assessment, investment strategies, and fraud detection, while in transportation, trustworthy decision-making algorithms can contribute to safe and efficient navigation and logistics. To advance this mission, our work on the GRD algorithm represents a significant step forward. GRD explores facilitating policy learning in the presence of delayed rewards. By decomposing the overall return into Markovian rewards, we provide a clearer understanding of the contribution made by each state-action pair. Furthermore, we go beyond simply explaining the rewards and delve into the causal view of reward generation. This approach allows us to provide interpretable explanations of how Markovian rewards are generated, enabling a more transparent decision-making process. This interpretability is vital for building trust with human users who may need to understand and validate the decisions made by the algorithm. Moreover, with an interpretable reward function, we can readily incorporate additional restrictions, such as security constraints, into the decision-making process. This flexibility allows us to tailor the algorithm to specific needs and requirements, further enhancing its trustworthiness. Additionally, since the principles and techniques we are exploring can be applied across a wide range of domains and industries, our collaboration with GRD not only contributes to the field of reinforcement learning but also has broader implications beyond robotics.

## B Proofs and Causal Background

### B.1 Markov and faithfulness assumptions

A directed acyclic graph (DAG), $\mathcal{G} = (V, E)$, can be deployed to represent a graphical criterion carrying out a set of conditions on the paths, where $V$ and $E$ denote the set of nodes and the set of directed edges, separately.

**Definition 1.** *(d-separation [50]). A set of nodes $Z \subseteq V$ blocks the path p if and only if (1) p contains a chain $i \rightarrow m \rightarrow j$ or a fork $i \leftarrow m \rightarrow j$ such that the middle node m is in $Z$, or (2) p contains a collider $i \rightarrow m \leftarrow j$ such that the middle node m is not in $Z$ and such that no descendant of m is in $Z$. Let $X$, $Y$ and $Z$ be disjunct sets of nodes. If and only if the set $Z$ blocks all paths from one node in $X$ to one node in $Y$, $Z$ is considered to d-separate $X$ from $Y$, denoting as $(X \perp_d Y \mid Z)$.*

**Definition 2.** *(Global Markov Condition [51, 50]). If, for any partition $(X, Y, Z)$, $X$ is d-separated from $Y$ given $Z$, i.e., $X \perp_d Y \mid Z$. Then the distribution $P$ over $V$ satisfies the global Markov condition on graph $G$, and can be factorizes as, $P(X, Y \mid Z) = P(X \mid Z)P(Y \mid Z)$. That is, $X$ is conditionally independent of $Y$ given $Z$, writing as $X \perp\!\!\!\perp Y \mid Z$.*

**Definition 3.** *(Faithfulness Assumption [51, 50]). The variables, which are not entailed by the Markov Condition, are not independent of each other.*

*Under the above assumptions, we can apply d-separation as a criterion to understand the conditional independencies from a given DAG $\mathcal{G}$. That is, for any disjoint subset of nodes $X, Y, Z \subseteq V$, $(X \perp\!\!\!\perp Y \mid Z)$ and $X \perp_d Y \mid Z$ are the necessary and sufficient condition of each other.*

### B.2 Proofs

**Proposition 1** (Identifiability). *Suppose the state $s_t$, action $a_t$, trajectory-wise long-term return R are observable while Markovian rewards $r_t$ are unobservable, and they form an MDP, as described in Eq. 2. Then under the global Markov condition and faithfulness assumption, the reward function g and the Markovian rewards $r_t$ are identifiable, as well as the causal structure that is characterized by binary masks $c^{\cdot \rightarrow \cdot}$ and $C^{\cdot \rightarrow \cdot}$ and the transition dynamics f.*

Below is the proof of Proposition 1. We begin by clarifying the assumptions we made and then provide the mathematical proof.

564 **Assumption** We assume that, $\epsilon_{s,i,t}$ and $\epsilon_{r,t}$ in Eq. 2 are i.i.d additive noise. From the weight-space
565 view of Gaussian Process [52], equivalently, the causal models for $s_{i,t+1}$ and $r_t$ can be represented as
566 follows, respectively,

$$s_{i,t+1} = f_i(s_t, a_t) + \epsilon_{s,i,t} = W_{i,f}^T \phi_f(s_t, a_t) + \epsilon_{s,i,t}, \tag{A1}$$

567

$$r_t = g(s_t, a_t) + \epsilon_{r,t} = W_g^T \phi_g(s_t, a_t) + \epsilon_{r,t}, \tag{A2}$$

568 where $\forall i \in [1, d^s]$, and $\phi_f$ and $\phi_g$ denote basis function sets.

569 Then we denote the variable set in the system by $V$, with $V = \{s_{1,t}, \ldots, s_{d^s,t}, a_{1,t}, \ldots, a_{d^a,t}, r_t\}_{t=1}^{T} \cup$
570 $R$, and the variables form a Bayesian network $\mathcal{G}$. Following AdaRL [23], there are possible edges
571 only from $s_{i,t-1} \in s_{t-1}$ to $s_{i',t} \in s_t$, from $a_{j,t-1} \in a_{t-1}$ to $s_{i',t} \in s_t$, from $s_{i,t} \in s_t$ to $r_t$, and from
572 $a_{j,t} \in a_t$ to $r_t$ in $\mathcal{G}$. In particular, the $r_t$s are unobserved, while $R = \sum_{t=1}^{T} \gamma^{t-1} o_t$ is observed. Thus
573 there are deterministic edges from each $r_t$ to $R$.

574 Below we omit the $\gamma$ for simplicity.

575 *Proof.* **Given trajectory-wise long-term return $R$, the binary masks, $c^{s \to r}$, $c^{a \to r}$ and Markovian**
576 **reward function $g$ and the rewards $r_t$ are identifiable.** Following the above assumption, we first
577 rewrite the function to calculate trajectory-wise long-term return in Eq. 2 as,

$$
\begin{aligned}
R &= \sum_{t=1}^{T} r_t \\
&= \sum_{t=1}^{T} \left[ W_g^T \phi_g(s_t, a_t) + \epsilon_{r,t} \right] \\
&= W_g^T \sum_{t=1}^{T} \phi_g(s_t, a_t) + \sum_{t=1}^{T} \epsilon_{r,t}.
\end{aligned}
\tag{A3}
$$

578 For simplicity, we replace the components in Eq. A3 by,

$$
\begin{aligned}
\zeta_g(X) &= \sum_{t=1}^{T} \phi_g(s_t, a_t), \\
E_r &= \sum_{t=1}^{T} \epsilon_{r,t},
\end{aligned}
\tag{A4}
$$

579 where $X := [s_t, a_t]_{t=1}^{T}$ representing the concatenation of the covariates $s_t$ and $a_t$ from $t = 1$ to $T$.
580 Consequently, we derive the following equation,

$$R = W_g^T \zeta_g(X) + E_r. \tag{A5}$$

581 Then we can obtain a closed-form solution of $W_g^T$ in Eq. A5 by modeling the dependencies between
582 the covariates $X_\tau$ and response variables $R_\tau$, where both are continuous. One classical approach
583 to finding such a solution involves minimizing the quadratic cost and incorporating a weight-decay
584 regularizer to prevent overfitting. Specifically, we define the cost function as,

$$C(W_g) = \frac{1}{2} \sum_{X_\tau, R_\tau \sim \mathcal{D}} (R_\tau - W_g^T \zeta_g(X_\tau))^2 + \frac{1}{2} \lambda \|W_g\|^2. \tag{A6}$$

585 where $\tau$ represents trajectories consisting of state-action pairs $X_\tau$ and long-term returns $R_\tau$, which
586 are sampled from the replay buffer $\mathcal{D}$. $\lambda$ is the weight-decay regularization parameter. To find the
587 closed-form solution, we differentiate the cost function with respect to $W_g$ and set the derivative to
588 zero:

$$\frac{\partial C(W_g)}{\partial W_g} = 0. \tag{A7}$$

Solving this equation will yield the closed-form solution for $W_g^T$, *i.e.*,

$$W_g = (\lambda I_d + \zeta_g \zeta_g^T)^{-1} \zeta_g R = \zeta_g (\zeta_g^T \zeta_g + \lambda I_n)^{-1} R \tag{A8}$$

Therefore, $W_g$, which indicates the causal structure and strength of the edge, can be identified from the observed data. In summary, given trajectory-wise long-term return $R$, the binary masks, $c^{s \to r}$, $c^{a \to r}$ and Markovian reward function $g$ and the rewards $r_t$ are identifiable.

**The binary masks, $C^{s \to s}$, $C^{a \to s}$ and the transition dynamics $f$ are identifiable** In a similar manner, based on the assumption and Eq. 2, we can rewrite Eq. A1 to,

$$s_{t+1} = W_{i,f}^T \phi_f(s_t) + \epsilon_{s,i,t}. \tag{A9}$$

To obtain a closed-form solution for $Wi$, $f^T$ in Equation A9, we can model the dependencies between the covariates $X_t$ and the response variables $st + 1$, both of which are continuous. The closed-form solution can be represented as:

$$C(W_{i,f}) = \frac{1}{2} \sum_{s_{i,t}, s_{i,t+1} \sim \mathcal{D}} (s_{i,t+1} - W_{i,f}^T \phi_{i,f}(s_t))^2 + \frac{1}{2} \lambda \|W_{i,f}\|^2. \tag{A10}$$

By taking derivatives of the cost function and setting them to zero, we can obtain the closed-form solution,

$$\begin{aligned} W_{i,f} &= (\lambda I_d + \phi_{i,f} \phi_{i,f}^T)^{-1} \phi_{i,f} s_{i,t+1} \\ &= \phi_{i,f} (\phi_{i,f}^T \phi_{i,f} + \lambda I_n)^{-1} s_{i,t+1}. \end{aligned} \tag{A11}$$

Therefore, $W_{i,f}$ can be identified from the observed data. This conclusion applies to all dimensions of the state. As a result, the $f$, which indicates the parent nodes of the $i$-dimension of the state, as well as the strength of the causal edge, are identifiable. In summary, the binary masks, $C^{s \to s}$, $C^{a \to s}$ and the transition dynamics $f$ are identifiable.

Considering the Markov condition and faithfulness assumption, we can conclude that for any pair of variables $V_i, V_j \in V$, $V_i$ and $V_j$ are not adjacent in the causal graph $\mathcal{G}$ if and only if they are conditionally independent given some subset of $\{V_l \mid l \neq i, l \neq j\}$. Additionally, since there are no instantaneous causal relationships and the direction of causality can be determined if an edge exists, the binary structural masks $c^{s \to r}$, $c^{a \to r}$, $C^{s \to s}$, and $C^{a \to s}$ defined over the set $V$ are identifiable with conditional independence relationships [26]. Consequently, the functions $f$ and $g$ in Equation 2 are also identifiable.

□

# C  Implementation Details

## C.1  Baselines

We compare our method against the following baselines,

- RRD (biased). This baseline utilizes a surrogate objective called randomized return decomposition loss for reducing the consumption of estimating the Markovian reward function. It applies Monte-Carlo sampling to get a biased estimation of the Mean Square Error (MSE) between the observed episodic reward and the sum of Markovian reward predictions in a sequence. We keep the same setting and hyper-parameters with its official implementation to reproduce the results, in which the policy module is optimized by soft actor-critic (SAC) algorithm [44].

- RRD (unbiased). This variant of RRD (biased) provides an unbiased estimation of MSE by sampling short sub-sequences. It offers a computationally efficient approach to optimize MSE. According to **(author?)** [14], RRD (biased) and RRD (unbiased) achieve state-of-the-art performance in episodic MuJoCo tasks.

- This baseline performs non-parametric uniform reward redistribution. At each time step, the proxy reward is set to the normalized value of the trajectory return. IRCR is a simple and efficient approach, and except for RRD, it achieves state-of-the-art performance in the literature. The implementation is from RRD [14].

**Algorithm 1** Learning the generative process and policy jointly.
___
1: Initialize: Environment $\mathcal{E}$, trajectory $\tau \leftarrow \emptyset$, buffer $\mathcal{D} \leftarrow \emptyset$
2: Initialize: Generative Model $\Phi_m := [\phi_{\text{cau}}, \phi_{\text{dyn}}, \phi_{\text{rew}}]$; Policy Model $\Phi_\pi$
3: **for** $i = 1, 2, \ldots, 3 \times 10^4$ **do**
4:     $\tau \leftarrow \emptyset$, reset $\mathcal{E}$
5:     **for** $n_{\text{step}} = 1, 2, \ldots, 100$ **do**
6:        sample data $\langle s_t, a_t, o_t \rangle$ from $\mathcal{E}$, and store them to trajectory $\tau$
7:        **if** $\mathcal{E}$ done **then**
8:           store trajectory $\tau = \{s_{1:T}, a_{1:T}, R\}$ to buffer $\mathcal{D}$, where $R = \sum_{i=1}^{T} \gamma^{t-1} o_t$
9:           $\tau \leftarrow \emptyset$, reset $\mathcal{E}$
10:        **end if**
11:        **for** $n_{\text{batch}} = 1, 2, \ldots, \text{train\_batches}$ **do**
12:           Sample $D_1$ consisting of $M$ trajectories from $\mathcal{D}$: $D_1 = \{\langle s_t^m, a_t^m \rangle \,|_{t=1}^{T}, R^m\} \,|_{m=1}^{M}$
13:           Sample $D_2$ consisting of $N$ samples from $\mathcal{D}$: $D_2 = \{s_{t_n}, a_{t_n}, s_{t_n+1}\} \,|_{n=1}^{N}$
14:           Sample binary masks by Gumbel-Softmax from $\phi_{\text{cau}}$: $\hat{c}^{s \to r}, \hat{c}^{a \to r}, \hat{C}^{s \to s}$ and $\hat{C}^{a \to s}$
15:           Sample binary masks deterministically from $\phi_{\text{cau}}$: $\tilde{c}^{s \to r}, \tilde{c}^{a \to r}$, and $\tilde{C}^{s \to s}$
16:           Calculate $\tilde{c}^{s \to \pi}$ based on $\tilde{c}^{s \to r}$ and $\tilde{C}^{s \to s}$
17:           Update $D_2$: $D_2 \leftarrow \{\tilde{c}^{s \to \pi} \odot s_{t_n}, a_{t_n}, s_{t_n+1}, \phi_{\text{rew}}(s_{t_n}, a_{t_n}, \tilde{c}^{s \to r}, \tilde{c}^{a \to r})\} \,|_{n=1}^{N}$
18:           Optimize $\phi_{\text{rew}}$ with $D_1$ (Using $\hat{c}^{s \to r}$ and $\hat{c}^{s \to r}$): $\phi_{\text{rew}} \leftarrow \phi_{rew} - \alpha \nabla_{\phi_{\text{rew}}} L_{\text{rew}}$ (Eq. 4)
19:           Optimize $\phi_{\text{dyn}}$ with $D_2$ (Using $\hat{C}^{s \to s}$ and $\hat{C}^{s \to s}$): $\phi_{\text{dyn}} \leftarrow \phi_{dyn} - \alpha \nabla_{\phi_{dyn}} L_{dyn}$ (Eq. 5)
20:           Optimize $\phi_{\text{cau}}$: $\phi_{\text{cau}} \leftarrow \phi_{\text{cau}} - \alpha \nabla_{\phi_{\text{cau}}} (L_{\text{sp}} + L_{\text{rew}} + L_{\text{dyn}})$ (Eq. 6)
21:           Optimize $\Phi_\pi$: $\Phi_\pi \leftarrow \Phi_\pi - \alpha \nabla_{\Phi_\pi} J_\pi$ (Eq. 8)
22:        **end for**
23:     **end for**
24: **end for**
___

## C.2   Detailed Generative Model

The parametric generative model $\Phi_m$ used in the MDP environment consists of three components: $\phi_{\text{cau}}$, $\phi_{\text{rew}}$, and $\phi_{\text{dyn}}$. We provide a detailed description of their model structures below.

$\phi_{\text{cau}}$ **for predicting the causal structure**   $\phi_{\text{cau}}$ comprises a set of free parameters without input. We divide $\phi_{\text{cau}}$ into four parts, each corresponding to the binary masks in Equation 2. Specifically, we have

- $\phi_{\text{cau}}^{s \to s} \in \mathbb{R}^{d^s \times d^s \times 2}$ for $C^{s \to s} \in \{0, 1\}^{d^s \times d^s}$,

- $\phi_{\text{cau}}^{a \to s} \in \mathbb{R}^{d^a \times d^s \times 2}$ for $C^{a \to s} \in \mathbb{R}^{d^a \times d^s}$,

- $\phi_{\text{cau}}^{s \to r} \in \mathbb{R}^{d^s \times 2}$ for $c^{s \to r} \in \mathbb{R}^{d^s}$,

- $\phi_{\text{cau}}^{a \to r} \in \mathbb{R}^{d^a \times 2}$ for $c^{a \to r} \in \mathbb{R}^{d^a}$.

Below we explain the shared workflows in $\phi_{\text{cau}}$ using the example of predicting the causal edge from the $i$-th dimension of state $s_{i,t}$ to the $j$-th dimension of the next state $s_{j,t+1}$, by part of the free parameters, $\phi_{\text{cau},i,j}^{s \to s}$.

For simplicity, we denote $\phi_{\text{cau},i,j}^{s \to s}$ as $\psi$. The shape of $\psi$ is now easy to be determined. That is $\psi \in \mathbb{R}^2$ and we write it as $\psi = [\psi_0, \psi_1]$. With this 2-element vector, we can characterize a Bernoulli distribution, where each element corresponds to the unnormalized probability of classifying the edge as existing ($\psi_0$) or not existing ($\psi_1$), respectively. Therefore, the probability of the causal edge existing from the $i$-th dimension of state $s_{i,t}$ to the $j$-th dimension of the next state $s_{j,t+1}$ can be calculated as:

$$P(C_{i,j}^{s \to s}) = \frac{\exp(\psi_0)}{\exp(\psi_0) + \exp(\psi_1)} \tag{A12}$$

**Obtain $\hat{C}_{i,j}^{s \to s}$ through Gumbel-Softmax sampling in the training phases.** During training, it is crucial to maintain the gradient flow for backpropagation. To achieve this, we sample the binary

| Layer# | 1 | 2 | 3 |
|---|---|---|---|
| $\phi_{\text{rew}}$ | FC256 | FC256 | FC1 |
| $\phi_{\text{dyn}}$ | FC256 | FC256 | FC9 |
| $\phi_{\pi}$ | FC256 | FC256 | FC2$d^a$ |
| $\phi_v$ | FC256 | FC256 | FC1 |

Table A1: The network structures of $\phi_{\text{rew}}$, $\phi_{\text{dyn}}$, $\phi_{\pi}$ and $\phi_v$. *FC256* denotes a fully-connected layer with an output size of 256. Each hidden layer is followed by an activation function, ReLU. $d^a$ is the number of dimensions of the action in a specific task.

values of $\hat{C}_{i,j}^{s \to s}$ by applying Gumbel-Softmax [41],

$$\hat{C}_{i,j}^{s \to s} = \text{GS}(\psi) \tag{A13}$$

where GS denotes the Gumbel-Softmax sampling, which allows us to obtain binary discrete samples from the Bernoulli distribution. By applying Gumbel Softmax sampling allows us to randomly sample from the Bernoulli distribution in a stochastic manner, rather than simply selecting the class with the highest probability. This introduces some randomness, enabling the model to explore the balance and uncertainty between different classifications more flexibly.

**Obtain $\tilde{C}_{i,j}^{s \to s}$ by deterministic sampling in the inference phases.** During inference, including data sampling and policy learning, we get the prediction of $C_{i,j}^{s \to s}$ through a deterministic sampling,

$$\tilde{C}_{i,j}^{s \to s} = \begin{cases} 1, \psi_0 \geq \psi_1 \\ 0, \psi_0 < \psi_1. \end{cases} \tag{A14}$$

This is a greedy sampling to avoid introducing randomness during the Gumble-Softmax sampling.

The above explanation of the workflow in $\phi_{\text{cau}}$ for predicting a single causal edge provides insight into the overall implementation of the entire module $\phi_{\text{cau}}$ and can be applicable for all the causal edges. Therefore, we can obtain $\hat{C}^{a \to s}$, $\hat{c}^{s \to r}$, $\hat{c}^{a \to r}$, $\tilde{c}^{s \to r}$ and $\tilde{c}^{a \to r}$, using similar procedures.

**$\phi_{\text{rew}}$ for predicting the Markovian rewards** $\phi_{\text{rew}}$ is a stacked fully-connected network, and the details of the network structure are provided in Table A1.

During training, the prediction of Markovian reward can be written as,

$$\hat{r} = \phi_{\text{rew}}(s_t, a_t, \hat{c}^{s \to r}, \hat{c}^{a \to r}) = \text{FCs}([\hat{c}^{s \to r} \odot s_t, \hat{c}^{a \to r} \odot a_t]), \tag{A15}$$

where $[\cdot, \cdot]$, $\odot$ denotes concatenation and element-wise multiply operations, respectively. FCs denotes the stacked fully-connected network. $\hat{c}^{s \to r}$ and $\hat{c}^{s \to r}$ are derived from $\phi_{\text{cau}}$ by Gumbel-Softmax.

During inference, including policy learning and data sampling, the predicted Markovian reward is

$$\tilde{r} = \phi_{\text{rew}}(s_t, a)_t, \tilde{c}^{s \to r}, \tilde{c}^{a \to r}) = \text{FCs}([\tilde{c}^{s \to r} \odot s_t, \tilde{c}^{a \to r} \odot a_t]), \tag{A16}$$

where $\hat{c}^{s \to r}$ and $\hat{c}^{s \to r}$ are derived from $\phi_{\text{cau}}$ greedily by deterministic sampling.

**$\phi_{\text{dyn}}$ for modeling the environment dynamics** In our experiment, we do not directly utilize $\phi_{\text{dyn}}$ in policy learning. Instead, this module serves as a bridge to optimize $\phi_{\text{cau}}^{s \to s}$ and $\phi_{\text{cau}}^{a \to s}$. Subsequently, $\phi_{\text{cau}}^{s \to s}$ can be utilized in the calculation of $\tilde{c}^{s \to \pi}$.

During training, we initially sample $\hat{C}^{s \to s}$ and $\hat{C}^{a \to s}$ using Gumbel-Softmax. The prediction for the $i$-th dimension of the next state can be represented as follows,

$$\hat{s}_{i,t} = \text{MDN}([\hat{C}_{\cdot,i}^{s \to s} \odot s_t, \hat{C}_{\cdot,i}^{a \to s} \odot a_t]), \tag{A17}$$

where $[\cdot, \cdot]$ denotes concatenation and MDN denotes the Mixture Density Network which outputs the means, variances, and probabilities for $N_{Gau}$ Gaussian cores. The parameters of MDN are shared across the predictions of different dimensions of the next state. We set $N_{cau} = 3$ in our experiments. More details about $\phi_{\text{dyn}}$ can be found in Table A1.

## C.3 Detailed Policy Model

Considering the specific requirements of the employed RL algorithm, Soft Actor-Critic (SAC), our Policy Model $\Phi_{\pi}$ comprises two components, the actor $\phi_{\pi}$ and the critic $\phi_v$. Detailed network structures for both components can be found in Table A1.

Table A2: The table of the hyper-parameters used in the experiments for GRD.

| Envs | $\lambda_1$ | $\lambda_2$ | $\lambda_3$ | $\lambda_4$ | $\lambda_5$ |
|---|---|---|---|---|---|
| *Ant* | $10^{-5}$ | $0$ | $10^{-7}$ | $10^{-8}$ | $10^{-8}$ |
| *HalfCheetah* | $10^{-5}$ | $10^{-5}$ | $10^{-5}$ | $10^{-6}$ | $10^{-5}$ |
| *Walker2d* | $10^{-5}$ | $10^{-5}$ | $10^{-6}$ | $10^{-6}$ | $10^{-7}$ |
| *Humanoid* | $10^{-5}$ | $10^{-8}$ | $10^{-5}$ | $10^{-7}$ | $10^{-8}$ |
| *Reacher* | $5 \times 10^{-7}$ | $10^{-8}$ | $10^{-8}$ | $10^{-8}$ | $10^{-8}$ |
| *Swimmer* | $10^{-7}$ | $10^{-9}$ | $10^{-9}$ | $0$ | $10^{-9}$ |
| *Hopper* | $10^{-6}$ | $10^{-6}$ | $10^{-6}$ | $10^{-7}$ | $10^{-6}$ |
| *HumanStandup* | $10^{-5}$ | $10^{-4}$ | $10^{-6}$ | $10^{-7}$ | $10^{-7}$ |

Table A3: The hyper-parameters.

| hyperparameters | value | hyperparameters | value |
|---|---|---|---|
| epochs | 3 | optimizer | Adam |
| cycles | 100 | learning rate | $3 \times 10^{-4}$ |
| iteration | 100 | $N$ | 256 |
| train_batches | 100 | $M$ | 4 |
| replay buffer size | $10^6$ | $\gamma$ | 1.00 |
| evaluation episodes | 10 | Polyak-averaging coefficient | 0.0005 |

## C.4 Training Process.

We follow the line of joint learning in **(author?)** [14], which avoids learning a return decomposition model in advance using data sampled by optimal or sub-optimal policies [13]. During each mini-batch training iteration, we sample two sets of data separately from the replay buffer $\mathcal{D}$:

- $D_1 = \{\langle s_t^m, a_t^m \rangle \mid_{t=1}^T, R^m\} \mid_{m=1}^M$ consists of $M$ trajectories. Provided with the trajectory-wise long-term returns $R^m \mid_{m=1}^M$, $D_1$ is utilized to optimize $\phi_{\text{cau}}^{s \to r}$, $\phi_{\text{cau}}^{a \to r}$ and $\phi_{\text{rew}}$, with $L_{\text{rew}}$.

- $D_2 = \{s_{t_n}, a_{t_n}, s_{t_n+1}\} \mid_{n=1}^N$ consists of $N$ state-action pairs. $D_2$ are used for policy optimization and optimize the parts of causal structure, $\phi_{\text{cau}}^{s \to s}$ and $\phi_{\text{cau}}^{a \to s}$, $\phi_{\text{dyn}}$. With such a $D_2$, GRD breaks the temporal cues in the training data to learn the policy and dynamics function.

Please refer to Algorithm 1 for a detailed training process.

## C.5 Hyper-Parameters.

The network is trained from scratch using the Adam optimizer, without any pre-training. The initial learning rate for both model estimation and policy learning is set to $3 \times 10^{-4}$. The hyperparameters for policy learning are shared across all tasks, with a discount factor of 1.00 and a Polyak-averaging coefficient of $5 \times 10^{-4}$. The target entropy is set to the negative value of the dimension of the robot action. To facilitate training, we utilize a replay buffer with a size of $1 \times 10^6$ time steps. The warmup size of the buffer for training is set to $1 \times 10^4$. The model is trained for 3 epochs, with each epoch consisting of 100 training cycles. In each cycle, we repeat the process of data collection and model training for 100 iterations. During each iteration, we collect data from 100 time steps of interaction with the MuJoCo simulation, which is then stored in the replay buffer. For training the $\phi_{\text{rew}}$, we sample 4 episodes, each containing $5 \times 10^3$ steps. As for policy learning and the optimization of $\phi_{\text{dyn}}$, we use data from 256 time steps. $\phi_{\text{cau}}$ is trained together with $\phi_{\text{rew}}$ and $\phi_{\text{dyn}}$. Validation is performed after every cycle, and the average metric is computed based on 10 test rollouts. The hyperparameters for learning the GRD model can be found in Table A2. All experiments were conducted on an HPC system equipped with 128 Intel Xeon processors operating at a clock speed of 2.2 GHz and 5 terabytes of memory.

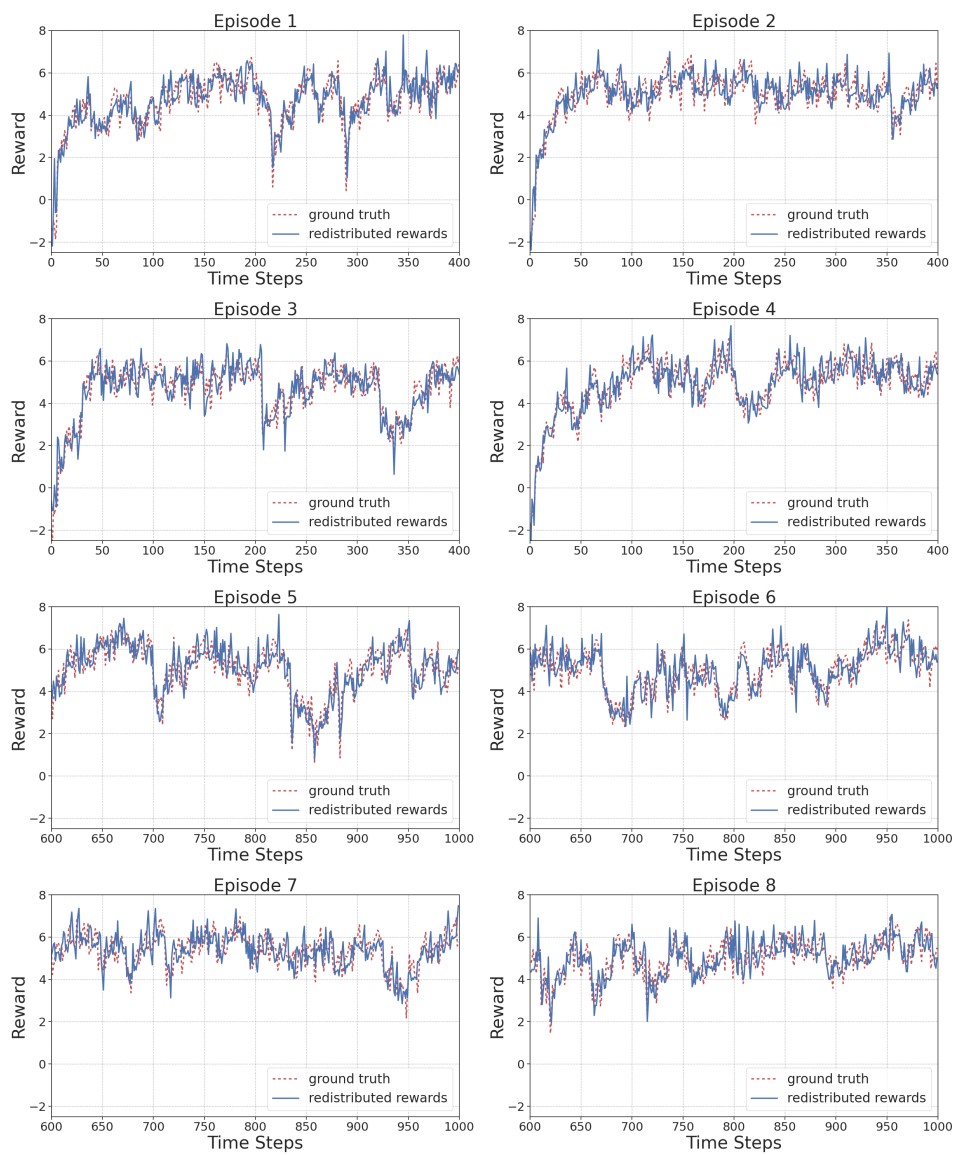

Figure A1: The visualization of redistributed rewards and grounded rewards in *Ant*. The results are produced by the GRD model trained after $1 \times 10^6$ steps. The redistributed rewards are shown in red, and the grounded rewards are shown in blue.

## D Visualization

As shown in Figure A1, we visualize the redistributed rewards in *Ant* by GRD, as well as the grounded rewards provided by the environment.