# OpenReview forum: "Interpretable Reward Redistribution in Reinforcement Learning: A Causal Approach"
_NeurIPS.cc/2023/Conference — NeurIPS 2023 poster_

### Official Review · Reviewer_24n7 · 2023-07-01

**Soundness:** 2 fair
**Presentation:** 2 fair
**Contribution:** 2 fair
**Rating:** 5
**Confidence:** 3

**Summary:**

This paper proposes a novel algorithm for return decomposition with causal treatment. To do reward redistribution, GRD uses factored representations to model the Markovian reward function and dynamics function.

**Strengths:**

The writing is clear and easy to follow. It is interesting to see the visualization in section 6.4, especially Fig. 4.

**Weaknesses:**

The technical contribution is somehow limited and the stronger experiments are expected.

**Questions:**

1 The technical contribution of the paper seems to be limited. The major contribution is the learned generative model. However, the proposed framework is similar to [27], which also uses factored MDP, learns a mask on the factors, and optimizes the model in a generative way. Is it correct to say GRD equals [27] adds markovian reward functions?

2 As for the experiments in Fig.3, GRD does not seem to outperform existing methods by a large margin. Except Halfcheetah, swimmer and humanoid stand up, GRD seems to be rather close to the baselines. Maybe it will be better to further test on some environment with more obvious performance gap.

3 It is interesting to see some horizontal lines in Fig. 4(a), where some dimensions have definite contribution to dimensions #27:54. I think it will be helpful to provide illustrations to correspond this to the mujoco ground truth dynamics, which proves that GRD indeed learns the true and interpretable casual structures.

---

> ### Author Rebuttal · Authors · 2023-08-10
>
> # Response to Reviewer 24n7
>
> Thank you for your positive support and constructive comments. We provide our point-wise response below.
>
> **Weakness 1:** The technical contribution is somehow limited and the stronger experiments are expected.
> > **Reply 1:** Thank you for your comments.
> > As complemented by Reviewer 6iFi, 1) "Interpretability: Having interpretable reward redistribution is an advantage over non-interpretable methods. This can be used to diagnose the reason for failures policy optimization." 2) "Reduces the state dimensionality: A very nice side effect of learning causal masks using a dynamics models is that a policy can be learned using very few features of the state. This leads to simpler policies, which could be more robust.", our contribution of providing an interpretable solution for delayed rewards is recognized by other reviewers, and we also reclarify our technical contribution in the Q1. As for the experiments, we provide additional experimental results on other RL training backbones (Figure 3 in attached PDF), on the tasks from Metaworld environment (Figure 4 in attached PDF) to demonstrate the state-of-the-art performance, as well as against noisy state to demonstrate the robustness of GRD. Please refer to the attached PDF.
>
> **Q1:** The technical contribution of the paper seems to be limited. The major contribution is the learned generative model. However, the proposed framework is similar to [27], which also uses factored MDP, learns a mask on the factors, and optimizes the model in a generative way. Is it correct to say GRD equals [27] adds markovian reward functions?
> > **A1:** Thanks for your questions. Our work differs from [27] in three ways. 1)Different task: [27] aims to identify the change factors across different domains and does not address the delayed rewards that we are interested in.  2) Identifiability: For GRD, without the given long-term return, the reward function and the related causal structure are not identifiable. For [27], the reward function is identifiable given the Markovian rewards, which are not observable in our setting. 3) Estimation method: due to the unobservable Markovian reward, we treat the long-term return as the causal effect of all the Markovian rewards within an episode and use the corresponding loss (Eq 4). Additionally, different from [27], we treat the existence of the causal edges as variables, leading to different losses for the minimal-edge assumption (likelihood in GRD and L1 loss in [27]) and different model structures (Gumbel-softmax for sampling in GRD). Overall, although we both utilize DBN, factored representation and model the causal structure as the binary masks, which are the common line for causal modeling [1][2], our work emphasizes the importance of learning an interpretable reward function and is the first to introduce causality into return decomposition.
>
>
>
> **Q2:** As for the experiments in Fig.3, GRD does not seem to outperform existing methods by a large margin. Except Halfcheetah, swimmer and humanoid stand up, GRD seems to be rather close to the baselines. Maybe it will be better to further test on some environment with more obvious performance gap.
> > **A2:** Thank you for the observation. We've extended our experiments to include the Meta-World environment, further showcasing our method's performance. Given the time constraints, we do not compare GRD with IRCR as it does not perform as well as RRD.  Besides this merit, GRD demonstrates 1) interpretability (our primary objective) and 2) robustness compared to the baseline methods. As an example of 2), upon introducing Gaussian noise to the $28\sim 111$ dimensions of states in Ant-v2, the performance of GRD does not decrease. Please refer to Figure 1 in the attached PDF for the results.
>
> **Q3:** It is interesting to see some horizontal lines in Fig. 4(a), where some dimensions have definite contribution to dimensions #27:54. I think it will be helpful to provide illustrations to correspond this to the mujoco ground truth dynamics, which proves that GRD indeed learns the true and interpretable causal structures.
> > **A3:** While the corresponding real causal structure would be helpful to verify the interpretability of our method, it is inaccessible. Therefore, we provide some evidence to underscore the reliability of the learned causal structure:
> > - there should not be an edge from the unused dimensions of the state variable to the other variables. (L334-L336)
> > - the edges from different dimensions of $\boldsymbol a$ to $r$ always exist, corresponding with the reward design of penalising the robot if it takes actions that are too large, measured by the sum of the values of all the action dimensions. (L340-342)
> >
> > Apart from above, we explain why the learned redundant edges (horizontal lines) do not impact the policy learning in L336-340: there is no edge from the  $28\sim 111$ dimensions of the next state to the reward, i.e., these dimensions do not exist in the identified compact representation, thus having no influence on policy learning.
> >
>
> **Reference**
>
> [1] Biwei Huang, Fan Feng, Chaochao Lu, Sara Magliacane, and Kun Zhang. Adarl: What, where, and how to adapt in transfer reinforcement learning. In International Conference on Learning Representations, 2021.
>
> [2] Huang, B., Lu, C., Leqi, L., Hernández-Lobato, J. M., Glymour, C., Schölkopf, B., & Zhang, K. (2022, June). Action-sufficient state representation learning for control with structural constraints. In International Conference on Machine Learning (pp. 9260-9279). PMLR.

---

> > ### Comment · Reviewer_24n7 · 2023-08-16
> >
> > I thank the authors for the clarifications and the efforts put into the rebuttal. I believe the authors have addressed most of the concerns. I will raise the score and please make the above modifications in the revised paper.

---

> > > ### Author Response · Authors · 2023-08-16
> > > **Response to Reviewer 24n7**
> > >
> > > Thank you for your positive feedback and recognition of our work. We will include the above modifications in the revised paper.

---

### Official Review · Reviewer_EmbG · 2023-07-03

**Soundness:** 3 good
**Presentation:** 3 good
**Contribution:** 3 good
**Rating:** 5
**Confidence:** 4

**Summary:**

This study introduces a novel approach, termed Generative Return Decomposition (GRD), to address a key challenge in reinforcement learning: identifying the state-action pairs that contribute to future, delayed rewards. While many methods redistribute rewards in a non-transparent manner, GRD offers a clear return decomposition by explicitly modeling the contributions of states and actions from a causal perspective.

GRD works by first recognizing unobservable Markovian rewards and causal relations in the data generation process. Then, it leverages these to create a compact representation for policy training over the agent's most favorable state-space subset. The researchers provide theoretical proof of the identifiability of the Markovian reward function and underlying causal structure and models. Experimental data also reveal GRD's superior performance and interpretability compared to other methods. However, some limitations exist due to the assumptions made, such as the stationary nature of the reward function, which may not be applicable in dynamic or online RL scenarios.

**Strengths:**

The paper excels in presenting Generative Return Decomposition (GRD), an innovative method that improves interpretability in reinforcement learning. GRD successfully addresses the identification of impactful state-action pairs for future rewards. Its effectiveness is supported by both theoretical evidence and practical experiments, demonstrating its superior performance over other existing methods. Moreover, the paper demonstrates a high degree of clarity and coherence, enabling smooth comprehension. Additionally, the explicit description of assumptions contributes to a more profound comprehension of the inherent strengths and weaknesses of the study.

**Weaknesses:**

- The quality of the text in Figure 1 could be improved by removing the shadow around the text. The same applies to Figure 2.
- Minor typos and errors that need to be edited for the next version of the paper. E.g. in line 172: "provide" -> "provides"
- In Section 5, only one policy is considered, which is SAC. How about having the experiments run based on another policy optimization algorithm? What would be the differences in performance and results?
- Regarding the last paragraph of Section 5, there are two possible scenarios to train the agent.
    - First, the generative model is learned while the policy is being updated, in an end-to-end paradigm.
    - Second, the generative model is first trained (and stays fixed thereafter), then the policy begins to be optimized.

    In either case, how would that affect the policy training and performance? And how the insights from the GRD interpretations would be changed?

- In the experiment section, the visualizations for the learned causal structure are only provided for Ant. Please provide the same type of analysis for other environments.

- Considering Figure 4, having GRD, how the agent's robustness and generalizability would be affected? For example, consider the case where there are some anomalies injected into the agent and environment interaction, more specifically changing some values from the state-space. If such anomalies target the features that are less important to the agent, then its performance should not be affected that much, right? If so, could you provide some results in this regard?

**Questions:**

See Weaknesses.

**Limitations:**

See Weaknesses.

---

> ### Author Rebuttal · Authors · 2023-08-10
>
> # Response to Reviewer EmbG
>
> We thank the reviewer for the comments. Below please see our responses as well as clarifications.
>
> **Weakness 1:** The quality of the text in Figure 1 could be improved by removing the shadow around the text. The same applies to Figure 2.
> > **Reply 1:** Thank you for your suggestion. We will revise the figures in the future version
>
> **Weakness 2:** Minor typos and errors that need to be edited for the next version of the paper. E.g. in line 172: "provide" -> "provides"
> > **Replay 2** Thank you for the comments. We will revise in the future version.
>
> **Weakness 3:** In Section 5, only one policy is considered, which is SAC. How about having the experiments run based on another policy optimization algorithm? What would be the differences in performance and results?
> > **Reply 3:** We provide results training with DDPG and TD3 in Figure 3 in the attached PDF. As the experimental result shows, on the tasks of *HalfCheetah*, GRD consistently outperforms the baseline methods, *RRD-Bias*, *RRD-Unbias*, and *IRCR*, which are modified to run based on the same policy optimization algorithm, DDPG and TD3. We also provide results of *None*, which utilizes the observed delayed reward for policy learning directly.
>
> **Weakness 4:** Regarding the last paragraph of Section 5, there are two possible scenarios to train the agent.
> First, the generative model is learned while the policy is being updated, in an end-to-end paradigm.
> Second, the generative model is first trained (and stays fixed thereafter), then the policy begins to be optimized.
> In either case, how would that affect the policy training and performance? And how the insights from the GRD interpretations would be changed?
> > **Reply 4:** Although these two lines are possible for learning the reward model, we follow the first line and enjoy the following advantages:
> > 1) We can use the on-training policy to collect data rather than the random policy, which would collect data without diversity.
> > 2) Apart from that, it is more data-efficient to optimize simultaneously since it avoids collecting data for learning the generative model and the policy separately.
> >
> > As for the insight of GRD, it will not be changed, since GRD models the generative process of the environment, which is totally the same for the two different lines.
>
> **Weakness 5:** In the experiment section, the visualizations for the learned causal structure are only provided for Ant. Please provide the same type of analysis for other environments.
> > **Reply 5:** Thank you for pointing that out. We provide the visualization of learned causal structure in *Swimmer-v2*, as shown in Figure 5 in the attached PDF, and more results for other environments will be presented in the future version.
> > Since the grounded causal structure is not accessible, we verify the reasonability of the learned causal structure by some observations:
> > - All the edges from different dimensions of $\boldsymbol a$ to $r$ always exist, as shown in Figure 5 (d): *Swimmer-v2* shares the same characteristic that the edges from different dimensions of $\boldsymbol a$ to $r$ always exist, corresponding with the reward design of penalizing the swimmer if it takes actions that are too large, measured by the sum of the values of all the action dimensions.
> > - According to Figure 5 (b), the first dimension of action (Torque applied on the first rotor) has an impact on the last three dimensions of state (angular velocity of front tip, angular velocity of first rotor, second rotor), which is corresponding with the intuitive that the part that connects to first rotor should be impacted by the first dimension of action. We can get a similar observation for the second action dimension from Figure 5.
> > - We can observe that all the state dimensions are learned to be connected to the reward; the possible explanation is that in the swimmer robot, any changes of the front tip, or two rotors will impact the position of the robot, potentially influencing the reward.
>
>
> **Weakness 6:** Considering Figure 4, having GRD, how the agent's robustness and generalizability would be affected? For example, consider the case where there are some anomalies injected into the agent and environment interaction, more specifically changing some values from the state-space. If such anomalies target the features that are less important to the agent, then its performance should not be affected that much, right? If so, could you provide some results in this regard?
> > **Reply 6:** Sure, we provide the results with certain noisy states at the insignificant dimensions of the state, as shown in Figure 1 in the attached PDF. To mimic the anomalies, we introduced independent Gaussian noise (mean: $0$, std: $0\sim 1$) to the dimensions ranging from $28$ to $111$ while evaluating the policy. According to Figure 1, unlike the baseline methods, GRD is unaffected by the injected noises, demonstrating that GRD is more robust than others. That is because the insignificant dimensions are not in the compact representation, which serves as the input of policy.

---

> > ### Comment · Reviewer_EmbG · 2023-08-12
> > **Response to Authors**
> >
> > Thanks for your thorough response. In your response, you referred to the "attached PDF" a few times. By that, do you mean the submitted paper?

---

> > > ### Author Response · Authors · 2023-08-12
> > > **Attached PDF**
> > >
> > > It is attached on the top of this page, which includes additional results and illustrations. Please kindly let us know if this help address your concerns. We are happy to answer any further questions should you have.
> > >
> > > Link: https://openreview.net/forum?id=w7TyuWhGZP&noteId=Mqm5eQDyt3.

---

> > > ### Author Response · Authors · 2023-08-19
> > > **Response to Reviewer EmbG**
> > >
> > > We hope that we already solve all of your concerns in the response and attached PDF. If you have any additional questions, we are more than willing to provide more clarification. Thank you once again for your time and patience.

---

### Official Review · Reviewer_ynzt · 2023-07-05

**Soundness:** 2 fair
**Presentation:** 2 fair
**Contribution:** 2 fair
**Rating:** 4
**Confidence:** 2

**Summary:**

Delayed reward in reinforcement learning is the major challenge in reinforcement learning. The return distribution technique is the direct way to resolve this issue while preserving policy. The existing works redistribute the returns in an uninterpretable manner. In this regard, this paper proposes a GRD which generates the Markovian rewards in delayed reward scenarios. GRD first checks the casual relations of state and actions and from a compact representation using causal generative model. The experiment results show that GRD outperforms the baselines and helps visualization.

**Strengths:**

* Experiment results seem promising.


**Weaknesses:**

* Explanation on line 345-352 is not sufficient and hard to understand. This experiment section is very important since authors insist that GRD give the interpretable structure of reward.
* The methods have to construct casual inference which is only possible when all the states are exactly defined. If the number of states explodes, the parameters to learn casual structure would explode. If states and actions are given, we can construct the casual structure without learning with parameters.

Minor
* Equations are too messy and hard to understand. Use under bracket in equation 6.
* Notations are not familiar. It is hard to understand C^{\cdot -> \cdot}, d^a , d^s (?). The authors have to redefine all the variables step-by-step to improve the presentation of this paper.
* The arrow size is not consistent in Figure 2. The arrow to \hat{r}_3 is narrower. Also, arrow directions which come from R are weird.

**Questions:**

* I could not understand the experiment results in l345-352.

**Limitations:**

Yes

---

> ### Author Rebuttal · Authors · 2023-08-10
>
> # Response to Reviewer ynzt
>
>
> Thanks for your constructive feedback. We provide a point-to-point response below.
>
> **Weakness 1:** Explanation on line 345-352 is not sufficient and hard to understand. This experiment section is very important since authors insist that GRD give the interpretable structure of reward.
>
> > **Reply 1:** We appreciate your observation, but there seems to be a misunderstanding. The experimental result (Line 345 - 352) aims to demonstrate the accuracy of predicting Markovian reward by our learned model, rather than the interpretable structure of the learned reward function (presented in L327 - 344).  We visualize the comparison of decomposed rewards and the ground truth rewards to demonstrate the accuracy of Markovian reward prediction by GRD. As shown in Figure 5, the blue lines, representing redistributed rewards, consistently align with the corresponding red lines, which are the ground truth. This visualization demonstrates that GRD indeed distinguishes the state action pairs with less contribution to the long-term return (episodic reward) from those with more contributions.  We will revise this section in the future version to provide a clearer explanation.
>
> **Weakness 2:** The methods have to construct causal inference which is only possible when all the states are exactly defined. If the number of states explodes, the parameters to learn causal structure would explode. If states and actions are given, we can construct the causal structure without learning with parameters.
> > **Reply 2:** Thanks for the question. This work focuses mainly on state-based tasks. 1) For the setting where states are not defined such as image-like input, our work can be applied by employed in the latent state space, which requires learning a latent vector representation, as AdaRL [1] does.  2) For a large number of dimensions of state, this would increase the complexity of the environment dynamics and thus requires more efficient causal discovery methods. 3) As an advantage, by learning the causal structure, we can constrain the optimization of the models over a small subspace of state and action, resulting in a lower requirement of parameters of the neural network. 4) Yes, it is possible to construct causal structures without learning the models, but it requires additional diverse data for causal discovery. Overall, the main focus of this work is to address the delayed rewards by interpretable return decomposition, therefore we consider the basic causal discovery algorithm.
>
>
> **Weakness 3:** Equations are too messy and hard to understand. Use under bracket in equation 6.
>
> > **Reply 3:** Eq. 6 is introduced to regulate the sparsity of learned causal structure to avoid trivial solutions. It is achieved by optimizing the parameters towards the direction of the nonexistence of the causal edge. Below, we revise Eq. 6 and hope to provide better delineation and understanding.
> >
> > Let $D_i(\boldsymbol{x})=\log P(\boldsymbol{x} _i)$, where $P(\boldsymbol{x}_i)$ is the possibility that the edge $\boldsymbol{x} _i$ exists, . Minimizing $D _i(\boldsymbol{s})$ prevents the causal edge from existing. Then Eq. 6 is,
> > \begin{array}{ll}
> L _{\text{sp}}(\phi _{\text{cau}}) =
> \underbrace{\lambda _1 \sum _i D _i(\boldsymbol{c}^{\boldsymbol s\rightarrow r})} _{\text{state-to-reward}} +
> \underbrace{\lambda_2 \sum _i D _i(\boldsymbol{c}^{\boldsymbol a\rightarrow r})} _{\text{action-to-reward}} + \\
> \underbrace{\lambda _3 \sum _{j \ne i} D _{i, j}(\boldsymbol{C}^{\boldsymbol s\rightarrow \boldsymbol s})} _{\text{state-to-state (excluding self-connections)}} +
> \underbrace{\lambda _4 \sum _{j = i} D _{i, j}(\boldsymbol{C}^{\boldsymbol s\rightarrow \boldsymbol s})} _{\text{state-to-state (self-connections)}} +
> \underbrace{\lambda_5 \sum _{j,i} D _{i, j}(\boldsymbol{C}^{\boldsymbol a\rightarrow \boldsymbol s})} _{\text{action-to-state}}.
> \end{array}
> >
> > These five terms are responsible for the sparsity of the causal structures of state-to-reward， action-to-reward， state-to-state (excluding self-connections)， state-to-state (self connections) and action-to-state, separately. Here self-connection represents the causal edge from $\boldsymbol{s} _{i, t}$ to $\boldsymbol{s} _{i, t+1}$.
>
> **Weakness 4:** Notations are not familiar. It is hard to understand $C^{\cdot \rightarrow \cdot}$, $d^a$ , $d^s$ (?). The authors have to redefine all the variables step-by-step to improve the presentation of this paper.
>
> > **Reply 4:** We genuinely appreciate your feedback on this matter, and we will revisit and clarify the symbols in our paper. Here we explain the mentioned notations:
> > - $\boldsymbol{C}^{\cdot \rightarrow \cdot}$ denotes all the causal masks, *i.e.*, $\boldsymbol{C}^{\cdot \rightarrow \cdot}:=[\boldsymbol{C}^{\boldsymbol{s}\rightarrow \boldsymbol{s}}, \boldsymbol{C}^{\boldsymbol{a} \rightarrow \boldsymbol{s}}, \boldsymbol{c}^{\boldsymbol{s} \rightarrow r}, \boldsymbol{c}^{\boldsymbol{a} \rightarrow r}]$;
> > - $d^\boldsymbol{s}$ denotes the number of dimensions of state;
> > - $d^\boldsymbol{a}$ denotes the number of dimensions of action.
>
>
> **Weakness 5:** The arrow size is not consistent in Figure 2. The arrow to $\hat{r}_3$ is narrower. Also, arrow directions which come from R are weird.
>
> > **Reply 5:** Thank you for pointing this out. We will revise and ensure consistency in the size and direction of the arrows in Figure 2.
>
>
> **Reference**
>
> [1] Biwei Huang, Fan Feng, Chaochao Lu, Sara Magliacane, and Kun Zhang. Adarl: What, where, and how to adapt in transfer reinforcement learning. In International Conference on Learning Representations, 2021.

---

> ### Author Response · Authors · 2023-08-19
> **Response to Reviewer ynzt**
>
> We sincerely hope that we have already solved all your concerns. We are also happy to solve any further questions.

---

### Official Review · Reviewer_6iFi · 2023-07-07

**Soundness:** 3 good
**Presentation:** 2 fair
**Contribution:** 4 excellent
**Rating:** 7
**Confidence:** 5

**Summary:**

The paper introduces a new algorithm called Generative Return Decomposition (GRD) for return decomposition with causal treatment. GRD addresses the problem by modeling causal relationships among variables, providing advantages over flat representations. It specifies each state and action as a combination of constituent variables and considers causal relationships within the system. The algorithm utilizes a factored representation similar to Factored MDP, enabling the formation and identification of the Markovian reward function based on causality. Unlike previous approaches, GRD uses a graphical representation to determine the contribution of each dimension of state and action to the Markovian reward. It also explains and models the observed delayed return as a causal effect of the unobserved Markovian reward sequence. The framework of GRD visualizes the causal relationships among environmental variables. The paper proves the identifiability of the underlying generative process and introduces a component-wise learning approach for recovering the causal generative process and redistributing rewards. The learned parameters provide a minimal sufficient representation for policy training, aiding in the effectiveness and stability of policy learning. The main contributions of the paper include the reformulation of return decomposition with a graphical representation, the introduction of GRD for learning the causal generative process, and empirical experiments demonstrating the method's superiority over state-of-the-art approaches in robot tasks with sparse rewards.

**Strengths:**

1 - Interpretability: Having interpretable reward redistribution is an advantage over non-interpretable methods. This can be used to diagnose the reason for failures policy optimization.

2 - Reduces the state dimensionality: A very nice side effect of learning causal masks using a dynamics models is that a policy can be learned using very few features of the state. This leads to simpler policies, which could be more robust.


**Weaknesses:**

1 - Writing: The paper needs a lot of work in explaining the method. Especially section 4 and section 5.1. A figure showing how the causal masks are applied would be a good idea. I am willing to improve my score, if the method explanation is improved.

2 - Experiments: The experiments include only Mujoco tasks. It would be interesting to see how the method behaves on delayed reward Atari environments like Bowling.

Missing Related work:
[1] Modern hopfield networks for return decomposition for delayed rewards

**Questions:**

1 - How do you decide which trajectory to store in the memory for training?

2 - Why is the dynamics model needed? The reward redistribution should be possible without the dynamics model.

**Limitations:**

Yes, the limitations have been addressed.

---

> ### Author Rebuttal · Authors · 2023-08-10
>
> # Response to Reviewer 6iFi
>
> Thank you for your positive support for our paper! Below we provide a point-wise response to your concerns.
>
> **Weakness 1:** Writing: The paper needs a lot of work in explaining the method. Especially section 4 and section 5.1. A figure showing how the causal masks are applied would be a good idea. I am willing to improve my score, if the method explanation is improved.
> > **Reply 1:** Thank you for your comments. We will revise Sec 4 and Sec 5.1 in the future version to include more clear and detailed explanations, including a figure for the overall framework. Additionally, please refer to Figure 2 in the attached PDF, which illustrates how the causal masks are applied.
>
>
> **Weakness 2:** Experiments: The experiments include only Mujoco tasks. It would be interesting to see how the method behaves on delayed reward Atari environments like Bowling.
> > **Reply 2:** Thank you for your suggestion. Our method focuses on state-based environments, which Atari does not satisfy. However, to further showcase the applicability of GRD, we provide additional experimental results:
> > 1) on three tasks from Meta-World, *pick-place-v2*, *push-wall-v2* and *door-lock-v2* (Figure 4) to demonstrate better performance of GRD, compared with the baseline methods.
> > 2) with different RL backbones (Figure 2), *TD3*, *DDPG*, to show the consistency improvement of GRD on the *HalfCheetah-v2*.
> > 3) against the Gaussian noise with different standard deviations in the insignificant dimensions of state in *Ant-v2* to demonstrate the more robust performance compared with baselines. During the evaluation, the noise is inserted in the $28\sim 111$ dimensions, which are not used in *Ant-v2*. The performance of GRD is not affected due to correctly identifying the compact representation for policy learning.
>
> **Weakness 3:** Missing Related work: [1] Modern hopfield networks for return decomposition for delayed rewards.
> > **Reply 3:** Thank you for pointing this out. Hopfield-RUDDER improves RUDDER by the replacement of LSTM with a continuous modern Hopfield network, and further employs history compression to facilitate the detection and storage of the key events. However, it still follows the line of RUDDER and shares the drawback of lacking interpretability. We will incorporate the suggested paper in the future version.
>
>
>
> **Q1:** How do you decide which trajectory to store in the memory for training?
> > **A1:** We collect trajectories using the on-training policy and subsequently store them in the buffer. During model training, we uniformly sample data from this buffer.
>
> **Q2:** Why is the dynamics model needed? The reward redistribution should be possible without the dynamics model.
> > **A2:** It is true that reward redistribution doesn't necessitate a dynamics model. We construct an ablation version of GRD, GRD wo CR. It only relies on the learned causal structure and reward function to train policy, and does not perform as well as GRD. Incorporating the dynamics model benefits policy learning by helping identify a compact representation: through learning the dynamics model, we gain insights into the causal relationships that dictate the generation of the next state, *i.e.*, how the $\boldsymbol{s} _{t}$ determine the next state $\boldsymbol{s} _{t+1}$ . Such causal knowledge helps identify the minimal sufficient dimensions of the state for policy learning ($\boldsymbol{\tilde{c}} ^{\boldsymbol{s}\rightarrow \pi}$ in Eq. 7), which is called compact representation in our paper. Compact representation is associated with the learned causal structure for generating Markovian reward and varies with the learning of the reward function. Since the supervision signals for policy learning are generated by the learned reward function, the policy learning consistently be optimized within the smallest state space, resulting in a more efficient training procedure.

---

> > ### Comment · Reviewer_6iFi · 2023-08-16
> >
> > Thank you for your response. I am increasing my score by 1 point.

---

> > > ### Author Response · Authors · 2023-08-16
> > > **Response to Reviewer 6iFi**
> > >
> > > We appreciate the reviewer for the positive feedback and recognition of our work.

---

### Official Review · Reviewer_Xv6U · 2023-07-23

**Soundness:** 2 fair
**Presentation:** 2 fair
**Contribution:** 2 fair
**Rating:** 5
**Confidence:** 1

**Summary:**

The paper addresses a major challenge in reinforcement learning: identifying which state-action pairs contribute to delayed future rewards. They propose a solution called "Return Decomposition" that redistributes rewards from observed sequences while maintaining policy invariance. Unlike other methods, their approach explicitly models state and action contributions from a causal perspective, making it interpretable.

The authors introduce a framework called "Generative Return Decomposition (GRD)" for optimizing policies in scenarios with delayed rewards. GRD identifies unobservable Markovian rewards and causal relationships in the generative process. Using this causal generative model, GRD creates a compact representation to train policies efficiently.

The paper proves the identifiability of the unobservable Markovian reward function and the underlying causal structure and causal models. Experimental results show that their method outperforms existing techniques, and visualizations demonstrate its interpretability. The source code for their approach is publicly available.

**Strengths:**

  - The authors provide theoretical proof for the identifiability of the unobservable Markovian reward function and the underlying causal structure. This solidifies the theoretical foundation and robustness of the model.
  - The GRD method outperforms state-of-the-art methods in experimental results across a range of tasks. This demonstrates its practical effectiveness and application potential.
  - Visualization of the learned causal structure and decomposed rewards contributes to the interpretability aspect, a valued characteristic in contemporary machine learning.

**Weaknesses:**

I have no knowledge about reinforcement learning, and I don't understand why the system assigned me to review papers on this topic. Please disregard my review comments.

**Questions:**

I have no knowledge about reinforcement learning, and I don't understand why the system assigned me to review papers on this topic. Please disregard my review comments.

**Limitations:**

I have no knowledge about reinforcement learning, and I don't understand why the system assigned me to review papers on this topic. Please disregard my review comments.

---

> ### Author Rebuttal · Authors · 2023-08-10
>
> # Response to Reviewer Xv6U
>
> We appreciate you for reviewing our paper! Thank you for your positive support!

---

### Author Rebuttal · Authors · 2023-08-10

# Attached PDF

Thank you to all the reviewers for your invaluable insights and thoughtful feedback. Your expertise has greatly helped us refine and improve our work.

Here we provide five figures in the attached PDF as supplementary of our response:

- Figure 1: Evaluation with Gaussian noise.
- Figure 2: The illustration of using learnable masks to predict the next state.
- Figure 3: Learning curves on *HalfCheetch-v2* with different training backbone, DDPG and TD3.
- Figure 4: Learning curves on three tasks from *MetaWorld*, Door Lock, Push Wall, and Pick Place.
- Figure 5: Learned causal structure in *Swimmer-v2*.

---

### Decision · Program_Chairs · 2023-09-21

**Decision:**

Accept (poster)

**Comment:**

The reviews and discussion offered cautious to enough positive support. We encourage the authors to use the feedback to further improve the work into the final version.